# Associations between Diabetes Mellitus and Selected Cancers

**DOI:** 10.3390/ijms25137476

**Published:** 2024-07-08

**Authors:** Monika Pliszka, Leszek Szablewski

**Affiliations:** Chair and Department of General Biology and Parasitology, Medical University of Warsaw, Chałubińskiego Str. 5, 02-004 Warsaw, Poland; monika.pliszka@wum.edu.pl

**Keywords:** type 1 diabetes mellitus, type 2 diabetes mellitus, cancer, mechanisms of association

## Abstract

Cancer is one of the major causes of mortality and is the second leading cause of death. Diabetes mellitus is a serious and growing problem worldwide, and its prevalence continues to grow; it is the 12th leading cause of death. An association between diabetes mellitus and cancer has been suggested for more than 100 years. Diabetes is a common disease diagnosed among patients with cancer, and evidence indicates that approximately 8–18% of patients with cancer have diabetes, with investigations suggesting an association between diabetes and some particular cancers, increasing the risk for developing cancers such as pancreatic, liver, colon, breast, stomach, and a few others. Breast and colorectal cancers have increased from 20% to 30% and there is a 97% increased risk of intrahepatic cholangiocarcinoma or endometrial cancer. On the other hand, a number of cancers and cancer therapies increase the risk of diabetes mellitus. Complications due to diabetes in patients with cancer may influence the choice of cancer therapy. Unfortunately, the mechanisms of the associations between diabetes mellitus and cancer are still unknown. The aim of this review is to summarize the association of diabetes mellitus with selected cancers and update the evidence on the underlying mechanisms of this association.

## 1. Introduction

According to data obtained from the World Health Organization (WHO), in 2019, cancer was the first or second cause of death before the age of 70 in 112 of 183 countries [1,2], and was a leading cause of more than 18 million cases worldwide in 2018 [3]. According to the data from WHO, the number of global patients with cancer will increase from 14 million in 2022 to 22 million in 2032 [4].

Diabetes mellitus (DM), a metabolic disorder characterized by hyperglycemia, hyperinsulinemia, and/or insulin resistance, is most often divided into two types. Type 1 diabetes mellitus (T1DM) is due to the autoimmune destruction of insulin-producing pancreatic β-cells, leading to insulinopenia. Type 2 diabetes mellitus (T2DM) is characterized by insulin resistance and compensatory hyperinsulinemia, causing β-cells’ failure. About 95% of diabetic patients have T2DM [5]. Type 3cDM, also known as pancreoprivic diabetes, is due to various diseases of the exocrine pancreas [6]. DM may be developed secondarily to medication or from an underlying illness. T3cDM is secondary to the exocrine pancreas, including, for example, pancreatic carcinoma, pancreatitis, pancreatic trauma, pancreatic resection, etc. This type of DM is characterized by a severe deficiency of all pancreatic glucoregulatory hormones [6]. Note that Alzheimer’s disease is also named type 3 diabetes mellitus (T3DM) [7]. The prevalence of DM has increased by more than 4-fold since 1980 [8]. The world current prevalence of DM is estimated to be at more than 400 million cases. In the next twenty years, in 2045, this number will rise to 650 million, with the majority being T2DM [9]. Based on other data, the global number of diabetic patients was 422 million in 2014, and the number of people with diabetes will increase to at least 592 million [10]. An adult patient in the US diagnosed with diabetes at age 60 loses 5 years of their life to the disease [11]. Based on the results obtained from performed investigations, there are suggested associations of DM with DM-related diseases, including cancers [12,13]. For example, analyses suggest an association of DM with an increased risk of developing cancers, such as liver, pancreatic, endometrial, colorectal, and breast in post-menopausal women [14]. Patients with diabetes have higher cancer-related mortality. It was found that 8–18% of patients with cancer have DM. The association between T2DM and cancer was first reported more than 100 years ago, and for nearly 200 years, a clear association of DM and pancreatic cancer has been known [15]. DM, especially T2DM, is associated with a higher risk for many cancers: an increased risk from 20 to 30% for breast or colorectal cancer, and a 97% increased risk of intrahepatic cholangiocarcinoma or endometrial cancer [16,17].

Anticancer therapy may be associated with the induction of diabetes mellitus, insulin resistance, hyperglycemia, hyperinsulinemia, and other pathologies associated with DM. Most chemotherapeutic drugs influence the cell cycle or damage DNA, resulting in apoptosis in rapidly dividing cells. Tegafur-uracil (UFT), paclitaxel, and interferon alpha may develop fulminant T1DM [18,19,20]. Interferon alpha can damage the function of pancreatic β-cells due to induction cytokines and enhance their susceptibility to invasion by diabetogenic T cells [21]. Glucocorticoids, impairing pancreatic β-cell function and insulin sensitivity, may cause hyperglycemia or DM [22]. Cancer immunotherapy with used anti-programmed cell death protein 1 (PD-1) or anti-programmed cell death ligand-1 (PDL-1) antibodies may induce the development of T1DM and cancer patients [23]. Several other associations between anti-cancer therapy and the development of DM, hyperglycemia, hyperinsulinemia, and so on have been described [24]. On the other hand, antidiabetic therapy may induce the development of cancers. This association was observed in the case of insulin, insulin analog therapy [25], incretin-based therapy [26], SGLT-2 inhibitors [27], sulfonylureas [28], and thiazolidinediones [29].

There are suggested several mechanisms to explain the involvement of DM in the development of cancer. The roles of inflammation, hyperglycemia, hyperinsulinemia, and antidiabetic drugs have been proposed [30]. On the other hand, in the case of prostate cancer, DM plays a protective role [31]. However, T1DM and T2DM are involved in increased risks of cancers, but T2DM has a stronger link with cancer. As mentioned earlier, the mechanisms of the association of DM with the incidence of cancer are still unknown. These mechanisms need intensive investigations, because clearly the prevention of T2DM may reduce the risk of cancer.

This review summarizes the associations between DM and cancers and presents hypotheses about the mechanisms of the correlation of diabetes and cancer. 

## 2. Diabetes Mellitus and Cancer

DM is a growing health problem worldwide. It is a common cause of mortality [32], and it significantly increases mortality in cancer patients [33,34]. Epidemiological studies have revealed that about 26.9% of all people over 65 have been diagnosed with diabetes and 60% have cancer worldwide [35]. Overall, from 8% to 18% of diabetic patients are also cancer patients [36]. In cancer patients, DM may cause a more aggressive clinical course of neoplasm, increase its metastatic potential, and may cause the patient’s organism to be less resistant to cancer progression.

Most evidence shows that DM plays the role of a risk factor for cancer, but there are also findings that this relationship is bidirectional: cancer may stimulate the development of diabetes [37]. 

### 2.1. Type 1 Diabetes Mellitus and Cancer

T1DM is characterized by profound insulin deficiency caused by the autoimmune destruction of pancreatic β-cells. This disease requires the exogenous administration of insulin. Type 1 diabetes mellitus differs from Type 2 and Type 3c. The parameters which characterize T1DM are ketoacidosis (common), hyperglycemia (severe), hypoglycemia (common), peripheral insulin sensitivity (normal or increased), insulin levels (low), glucagon levels (normal or high), pancreatic peptide (PP) levels (normal or low), glucose-dependent insulinotropic polypeptide (GIP) levels (normal or low), glucagon-like peptide 1 (GLP1) levels (normal), and typical age of onset (childhood or adolescence) [6]. T1DM has not been linked with the increased risk of cancer. Knowledge about the risk of cancer in patients with T1DM is poor, and the association of T1DM with cancer is not well described [38]. 

As mentioned above, there are fewer studies on the association of T1DM with cancer. The results obtained from one cohort study revealed that T1DM was associated with an increased risk of stomach, cervical, and endometrial cancer [38]. The cancer risk for overall cancer patients with T1DM increased by 20% in comparison to patients without diabetes. The standardized incidence rate (SIR) and 95% confidence interval (CI) for patients with DM were 1.2 and 1.0–1.3, respectively. Diabetic patients had increased risks of cancers of the stomach (SIR 2.3; CI 1.1–4.1), cervix (SIR 1.6; CI 1.1–2.2), and endometrium (SIR 2.7; CI 1.4–4.7). Based on the obtained results, the authors suggested that T1DM is associated with a modest risk of specific cancers, however, they differ from those associated with T2DM [38,39,40,41,42,43]. Another study showed a 17% increased risk of cancer in patients with T1DM in comparison to patients with cancer without T1DM. In patients with DM, an increase in specific cancers, such as gastric cancer, squamous cell skin carcinoma, and leukemia [44], was detected. Very interesting results have been obtained in other studies [40]. A systematic review revealed that T1DM increases the risk of gastric cancer by 46%. This result was in contrast to results obtained in another study, showing that T1DM did not increase the risk of gastric cancer [45]. The authors of a previous study suggested that these differences in obtained results are due to the uses of different types of insulin analog. It was found that a dose-dependent increased risk of cancer is observed with glargine therapy [46]. Other factors are also postulated. T1DM is an autoimmune disease with several pathogeneses. Autoimmune disorders, such as autoimmune gastritis and pernicious anemia, are more likely to occur in patients with T1DM [47]. It is suggested that comorbid autoimmune disturbances may be associated with the risk of gastric cancer. The second factor that may influence gastric cancer risk is *Helicobacter pylori* infection [40]. This bacterium is the most common risk factor for gastric cancer. A higher risk of *H. pylori* infection is observed in children with T1DM [48]. Observations performed for non-sex-specific cancers showed that the estimated hazard ratio (HR) for overall cancers was 1.01 among men and 1.07 among women in comparison with the general population [49]. Results obtained from other research showed a significantly increased risk of cancer incidence such as liver, pancreas, kidney, esophagus, stomach, lung, thyroid, squamous cell carcinoma, and leukemia in patients of both sexes with T1DM [50]. Non-Hodgkin’s lymphoma and colon cancer were evaluated in men with T1DM [49]. In women with T1DM, the incidence of the ovary, esophagus, endometrium, vulva, vagina, and thyroid cancer was significantly increased as compared to women without DM [49]. The incidence of prostate and testis cancer was significantly decreased in diabetic men in comparison with the general population [42,49]. In women with T1DM a decreased risk of cancer was observed in the case of breast cancer, melanoma, and Hodgkin’s lymphoma [49,51]. Also found was no increased risk of cancer overall in patients with T1DM, while there was observed a heightened risk of ovarian cancer incidence and mortality [52]. Because these obtained results are different and sometimes controversial, further studies are needed to determine an association of T1DM with the increased incidence and mortality of specific cancers [53]. 

### 2.2. Type 3c Diabetes Mellitus and Cancer

T3cDM, also known as pancreoprivic or pancreatogenic diabetes, which is secondary to pancreatic diseases, is due to various diseases of the exocrine pancreas [54], such as pancreatic carcinoma, acute and chronic pancreatitis, cystic fibrosis, pancreatectomy, rare genetic, and several others [50]. The characteristic parameters for it are ketoacidosis (rare), hyperglycemia (mild), hypoglycemia (common), peripheral insulin sensitivity (increased), hepatic insulin sensitivity (decreased), insulin levels (low), glucagon levels (low), PP levels (low), GIP levels (low), GLP1 levels (normal or high), and typical age of onset (any) [6]. The prevalence of diabetes in patients with diseases of the exocrine pancreas is approximately 0.11% [55], and approximately 9.2% of diabetic patients have T3cDM [56]. T3cDM is a major subset of the total population of DM, and is associated with the highest risk of pancreatic cancer (PC), especially in individuals with T3cDM secondary to chronic pancreatitis (CP) [57,58]. It was found that 4–5% of patients with any form of CP develop pancreatic cancer over the course of 20 years and the risk of this incidence is 10–20 times greater in comparison with the global population [59]. A performed case–control study revealed an increased risk of PC, relative risk (RR = 4.7), in patients with pancreatitis [60]. A cohort study in Taiwan showed with CP hazard ratios (HR = 19.40) that factors that significantly predicted PC were gallstones (HR = 2.56) and hepatitis C infection (HR = 3.08). Patients with DM and CP had an elevated risk of developing PC (HR = 33.52) compared with patients without these comorbidities [61]. The most common cause of T3cDM is CP. Chronic pancreatitis increases the risk of pancreatic cancer 10- to 20-fold [50]. On the other hand, CP and DM increase the risk of pancreatic cancer development 33-fold. 

In North America, T3cDM is detected in all diabetic patients. Similar observations revealed that on the Indian and southeast Asian subcontinents, this disease affects as many as 15–20% of diabetic patients. On these subcontinents, tropical or fibrocalcific pancreatitis is endemic [62]. In Germany, in 2000 diabetic patients, T3cDM was detected [63]. It was also shown that nearly half of patients with T3cDM had been previously misdiagnosed as having either T1DM (6%) or T2DM (40%) [56]. Another cohort study in Germany revealed that 78.5% of T3cDM patients had CP as a cause of their diabetes mellitus. In 500 patients with CP as an effect of alcoholism, T3cDM developed in 8.3% within 25 years of the diagnosis of CP [64].

### 2.3. Gestational Diabetes Mellitus and Cancer

Gestational diabetes mellitus (GDM) is diagnosed during pregnancy without previous diabetes, characterized by a degree of carbohydrate intolerance [65]. Obtained results have revealed that women with a history of GDM have an approximately 7.4 times higher risk of developing postpartum T2DM, in comparison with women who, during pregnancy, had a normal glucose tolerance [66]. Investigations of any association between GDM and cancer are very few [67,68,69,70]. An association of GDM with thyroid cancer was analyzed. A performed meta-analysis of cohort studies revealed that the thyroid cancer risk in women with GDM was (RR = 1.30; 95% CI = 1.17–1.43) and the risk of thyroid cancer development was increased by 30% in comparison with women without GDM. For women with T1DM and T2DM, the risk of thyroid cancer was 34% and 51%, respectively [71]. There are also studies on the effect of GDM on breast cancer (BC) incidence. Unfortunately, the results obtained are different, contrary, and controversial. Only one study reached statistical significance, where the author found that GDM reduces the risk of BC [72]. One study revealed a negative risk point estimate [73]. The pooled OR did not reveal any association between GDM and BC (OR = 1.06; 95% CI = 0.79–1.40). Three studies showed an increased risk point estimate [74,75,76]. 

### 2.4. Type 2 Diabetes Mellitus and Cancer

T2DM is a metabolic disease due to the insulin resistance of peripheral tissues and cells, resulting in increased blood glucose levels (hyperglycemia). Chronically, prolonged increased blood glucose levels may cause end organ damage, such as nephropathy or retinopathy. An elevated level of circulating glucose stimulates the increased secretion of insulin by pancreatic β-cells to obtain normoglycemia. Both hyperglycemia and hyperinsulinemia may be associated with the development of cancer [9,13,50]. The parameters which characterize T2DM are ketoacidosis (rare), hyperglycemia (usually mild), hypoglycemia (rare), peripheral insulin sensitivity (decreased), hepatic insulin sensitivity (normal or decreased), insulin levels (high), glucagon levels (normal or high), PP levels (high), GLP-1 (normal or low), and typical age of onset (adulthood) [6].

Several studies have revealed that T2DM is an independent risk factor that may influence risk cancers, such as hepatic, pancreatic, bladder, endometrial, colorectal, and breast cancer. There are several other associations between T2DM and an increased risk for the development of cancer, including kidney, glioma, melanoma, and gynecological cancer [17,50]. On the other hand, there are also contrary results showing that, in diabetic patients, T2DM has no effect on the risk of some cancers, such as brain, buccal cavity, bladder, laryngeal, and lung [77], as well as a decreased risk [78]. Based on meta-analyses, it was suggested that T2DM is linked with a 25–41% increased risk of mortality from any cancer [79,80,81], with controversial results which suggest a null association between T2DM and the risk of death due to some cancers [82,83]. Therefore, intense studies are needed for further clarification of this association. The potential link between T2DM and some types of cancer, as well as its mechanisms, has been discussed for decades [84]. 

#### 2.4.1. Pancreatic Cancer

Pancreatic cancer is one of the most lethal malignant diseases. Its five-year survival rate is less than 10% [31], and after surgical resection, this survival rate rises to about 20%. Note that only 20% of patients may be candidates for surgical resection [85]. In the United States, it is the tenth most common cancer and the fourth cause of death [86].

For nearly 200 years, a positive association of DM with PC [15] has been known. This is a problem, because epidemiological studies revealed that T2DM is a risk for PC, however, there are studies that suggested that new-onset diabetes is caused by PC [15,87,88,89]. A performed meta-analysis on the association of PC and T2DM revealed that long-term DM increases the risk of PC [90]. Patients with T2DM ≥ 2 years have a 1.5 to 1.7-fold increased risk of developing PC. The authors also investigated the relative risk (RR) of developing PC depending on the duration of T2DM. The results obtained showed that the RRs for diabetic patients with a duration of DM ≥ 2, ≥5, and >10 years were 1.64 (95% CI = 1.52–1.78), 1.58 (95% CI = 1.42–1.75), and 1.50 (95% CI = 1.28–1.75), respectively. This result may be associated with changes in lifestyle and/or antidiabetic medications [90]. Other research revealed that patients with new-onset T2DM had RR = 7.94 (95% CI = 4.70–12.55) for developing PC in comparison with patients without DM. Researchers also did not find associations between body mass index (BMI) and the risk of developing PC [91]. Results obtained by other authors confirmed the suggestion that T2DM increases the risk of developing PC [92,93], as well as observing an increased risk of pancreatic ductal adenocarcinoma in diabetic patients [94]. As in previous investigations, the researchers did not find an association between BMI and PC risks, while another meta-analysis revealed a twofold risk of PC in patients with DM (RR = 1.94; 95% CI = 1.66–2.27). It was also found that the PC risk is not dependent on sex, geographic location, BMI, or alcohol consumption. On the other hand, the obtained results revealed a negative correlation between the risk of PC with the duration of diabetes. The highest risk of PC was observed in patients with diabetes diagnosed < 1 year (RRs = 5.38), when the risk of PC then gradually decreased. A higher risk of PC was observed in patients with a duration of T2DM of 1–4 years (RRs = 1.95) compared to patients with a duration of diabetes of 5–9 years RRs = 1.49 [95]. A cohort study performed on African American and Latinos revealed the association between newly onset DM and pancreatic cancer. Newly onset diabetes is linked with a 2.3-fold increased risk of PC in comparison with long-term DM. According to the authors’ suggestion, the new onset of DM is a sign of pancreatic cancer [96]. More investigations showed an inverse duration-dependent risk of DM and PC. A remarkable rate of PC occurrence is seen in the first 2 years after the diagnosis of DM; then, the rate decreases as time goes by. The risk of pancreatic cancer decreases significantly in diabetic patients who suffer DM > 5 years [97,98]. Based on the obtained results, it is suggested that long-term T2DM is a risk factor for PC, whereas newly diagnosed DM is a sign of PC [31]. Between diabetes and PC, a clear association is observed. Unfortunately, there are difficulties to explain: considering the causes of the involvement of insulin resistance and hyperinsulinemia, elevated levels of insulin-like growth factor-1 (IGF-1) have been suggested. Islet cells may play an important role, because hyperactivity and the increased β-cell mass cause insulin over-secretion and insulin resistance. Animal studies revealed that the stimulation of islet cells’ proliferation increases carcinogenesis of pancreatic ductal cells, whereas the destruction of these cells by streptozotocin or alloxan decreases the carcinogenesis of these cells [99]. Interesting results were obtained in a study on the association between prediabetes and PC. At cancer diagnosis, the prevalence of prediabetes was 6% (95% CI = 5.3–6.8), while the prevalence of diabetes was 12.2% (95% CI = 11.2–13.3). One year after diagnosis, this prevalence was 16.6 (95% CI = 15.4–17.9) for prediabetes and 25.0% (95% CI = 23.6–26.4) for diabetes. In the next year, the prevalence of prediabetes was 21.2% (95% CI = 19.9–22.6) and the prevalence of diabetes was 32.6% (95% Cl = 31.1–34.2). The obtained results showed that patients with PC had the highest prevalence of DM (65.1%; 95% CI = 57.0–72.3) in comparison to patients with other cancers [100]. There are also other studies which confirm the association between T2DM and PC [101]. The onset of T2DM in 40% of patients with PC was also predominantly found [102]. Based on the obtained results, it is suggested that pancreatogenic diabetes, so-called T3cDM, should be distinguished from new-onset T2DM. This suggestion is based on obtained characteristic results, such as a negative family history of DM, recent weight loss > 2 kg, BMI < 25 kg/m^2^, and age ≥ 65 years [103].

#### 2.4.2. Primary Liver Cancer

Hepatocellular carcinoma (HCC) is the most common type of primary liver cancer [104]—the fifth most common cancer in men and seventh one in women. Its incidence is especially high in east Asia and Africa [105]. Performed studies revealed a strong association of T2DM and HCC, and DM as an independent risk factor for HCC. Of note, HCC is also associated with infections by hepatitis B virus (HBV), hepatitis C virus (HCV), aflatoxin exposure, and non-alcoholic fatty liver disease (NAFLD) [105,106]. The association between DM and HCC with a high prevalence of hepatitis virus infection was investigated. The obtained results revealed that the increased risk of HCC due to diabetes is more significant in HCV-negative patients than in HCV-positive patients [107]. In more than 70% of diabetic patients, NAFLD is diagnosed as being caused by insulin resistance [107,108], and individuals with diabetes are more susceptible to severe liver diseases, such as HCC. Another study revealed a two- to three-fold increased risk of HCC in patients with T2DM [109]. The increased incidence of HCC is well documented, however, the mechanisms of this association are still unclear [109]. A positive correlation between T2DM and HCC, biliary tract, and gallbladder [8] was observed. Results obtained in other studies also showed a significant positive correlation between T2DM and mortality due to cancers of the liver, gallbladder, and bile duct [82]. The associations of T2DM and hepatic cancer were confirmed in other studies [110,111]. In some patients, hepatic cancer can progress from simple steatosis to inflammation resulting in nonalcoholic steatohepatis (NASH). Among diabetic patients, the prevalence of NASH was estimated to be 22.2% [112]. NASH can be involved in the development of fibrosis cirrhosis and HCC [113]. It was found that NAFLD may contribute to an increased proportion of cryptogenic cirrhosis cases [114,115,116]. Cryptogenic cirrhosis is associated with 30–40% of HCC cases in Western societies [117]. Of note, HCC may arise from NASH without a preexisting cirrhosis stage, or even from NAFLD with mild or absent fibrosis [117,118]. A performed analysis of 162 cases of HCC revealed an involvement of NAFLD in HCC development in 24% of patients [119]. A meta-analysis also revealed an association between T2DM and the risk of cholangiocarcinoma (CC), including its intra- and extrahepatic location. In this investigation, the obtained results showed an increased risk of CC in diabetic patients: RR = 1.60 (95% CI = 1.38–1.87) [120]. The association of CC with T2DM may be caused by the increased formation of biliary stones, which is a risk factor for CC, while diabetes and insulin resistance independently promote the formation of gallstones [121].

#### 2.4.3. Esophageal Carcinoma

Esophageal carcinoma (EC) is the sixth most common cause of death [122]. It is also, as previously mentioned, a cancer associated with T2DM. According to a meta-analysis of 17 studies, DM increases the risk of cancer, however, at a modest level (SRR = 1.30; 95% CI 1.12–1.50) [123,124]. An analysis performed in 2011 revealed SRRs for diabetic men of 1.28 (95% CI = 1.10–1.49) and for diabetic women of 1.07 (95% CI = 0.71–1.62) [123]. There are two main histological subtypes of EC: squamous cell carcinoma (ESCC) and esophageal adenocarcinoma (EAC) [36]. For adenocarcinoma, there was a combined SRR = 2.12 (95% CI = 1.01–4.46). A rise in the incidence rates of EAC was also found, which is estimated to account for 30%-50% of all cases of EC. The authors found that twenty years ago, EAC comprised only 5% of esophageal malignancies [123]. 

#### 2.4.4. Gastric Cancer

Gastric cancer (GC) is the fifth most common type of cancer and the fourth cause of cancer mortality globally. The overall 5-year survival rate for GC is less than 20% in a few countries. However, meta-analyses have showed increased risks of several cancers associated with DM, and it has been discussed whether diabetes may be a potential risk for gastric cancer [40]. A meta-analysis of cohort studies revealed that the risk of GC was 14% higher in patients with T2DM in comparison with cancer patients without DM. The RRs for T2DM patients ranged from 1.11 (95% CI = 1.04–1.19) to 1.16 (95% CI = 1.08–1.24). Authors have found that the risk of gastric cancer was the highest one year after T2DM (RR = 1.29 was diagnosed; 95% CI = 1.09–1.51). Then, two and three years after the diagnosis of T2DM, the risk of GC increased nearly to the initial level (RR = 1.26; 95% CI = 1.01–1.56) [40]. Authors have analyzed results depending on the continent also. A significant correlation between T2DM and GC incidence was found in studies in Asia (RR = 1.14; 95% CI 1.04–1.26], but not studies in America, Australia, and Europe (RR = 1.12; 95% CI = 0.99–1.26). For more details, see [40]. A positive correlation between T2DM and GC was obtained in other studies (RR = 1.16; 95% CI = 1.01–1.33) [45]. Many studies revealed a positive correlation between T2DM and GC [39], as well as a dependence on sex: in female patients for GC (RR = 2.21; 95% CI = 1.92–2.55), and for male patients (RR = 1.78; 95% CI = 1.61–1.97) [110]. There are also some studies which showed that T2DM is not associated with GC. These results were taken from patients from China [125], Korea [126], Israel [127], the USA [128], Sweden [129], Italy [130], France [131], and the UK [132]. It is worth noting that a few studies showed a lower-risk inverse association of GC with T2DM. These results were obtained in patients from China [133], Israeli Arabs [134], and from Italy [135].

#### 2.4.5. Oral Cancer

DM is involved in various oral diseases, such as xerostomia, periodontal disease, oral candidiasis, etc. [136]. Also investigated is the association between T2DM and oral cancer. The obtained results revealed that diabetic patients have an increased risk of both precancerous lesions and oral cancer [137,138]. Unfortunately, there is no consensus about the mechanisms linked to these associations [139]. More recent research suggests the association of DM and oral cancer. Authors have found that the prevalence of glucose metabolism disorder, which may be due to diabetes, was significantly common among cancer patients (59.9% vs. 36.5%). The most common tumor types were squamous cell carcinoma (93.4%) and sublingual tumor location (35.5%). Authors suggest that not only is T2DM involved in oral cancer development, but smoking is also [136].

#### 2.4.6. Colorectal Cancer

Colorectal cancer (CRC) is the fourth most common cancer and the second cause of death in the USA. In low- and middle-income countries, the mortality rate caused by CRC is 33% [140]. It is the third most common cancer worldwide and the second reason for CRC-specific deaths [141]. In Poland, CRC is the second most common cause of mortality due to cancers [142]. Since 1980, the number of colon cancers has increased by about four times in men and nearly three times in women [142]. An association of T2DM and an elevated risk of CRC has been observed in many studies. Obtained results showed that CRC is closely associated with T2DM, and T2DM is an independent risk factor for CRC. In diabetic patients with CRC, a higher mortality has been found in comparison with nondiabetic patients with CRC [31,143]. Only a small risk of CRC was observed in female patients with diabetes, but a significantly increased risk in men with T2DM was found [144]. The results obtained from a performed cohort study demonstrated an increased risk of CRC in diabetic men (RR = 1.24; 95% CI = 1.08–1.44). In women with diabetes, there was no association with the risk of CRC (RR = 1.22; 95% CI = 1.04–1.45) [145]. Another Swedish study showed a 49% increased risk of CRC in diabetic men in comparison to men with CRC without T2DM [146]. Interesting results were obtained in another study in which the majority of participants were African American. A 47% increased risk of developing CRC was observed in patients with diagnosed diabetes as compared to patients with CRC but without T2DM. It was found also that this observed association was greater for patients without recent colonoscopy screenings and patients with a more recently diagnosed DM [147]. A population-based cohort study revealed a 1.26-fold increased risk of developing CRC in patients with T2DM (HR = 1.96; 95% CI = 1.18–1.33) [148]. The results obtained in another study revealed that diabetic patients were 30% more likely to develop CRC and 70% more likely to develop proximal colon cancer in comparison with individuals without T2DM. The effect of DM on developing CRC was seen to be pronounced in men under the age of 55 years [149]. In patients with CRC and diagnosed T2DM, a 12% increase in cancer-specific mortality (RR = 1.12; 95% CI = 1.01–1.24) [150] was reported, and decreases in overall survival of 18%, 19%, and 16% with colorectal, colon, and rectal cancer, respectively, as compared with patients without T2DM [151]. Investigations of 19 single-nucleotide-polymorphisms (SNPs) associated with T2DM showed that four SNPs were associated with a risk of CRC development and only one (KCNJ11) was associated with an increased risk of CRC (Odds ratio, OR = 1.18; 95% CI = 1.05–1.32) [152]. The results obtained in a Swedish nationwide cohort study showed that diabetic men and women reached the risk levels for 50-year-old individuals (0.40% and 0.41%, respectively) at about the age 45 instead of 50. This is nearly 5 years earlier than the general population. If these patients had an additional family history of CRC, they reached this risk threshold from 12 to 21 years earlier in comparison with the general population [153]. The risk of CRC is not dependent on sex, but the strongest influence on CRC development is the duration of T2DM (11–15 years) [154]. Performed studies have shown a positive correlation between a higher occurrence of adenomatous polyps which are detected at a younger age [155,156,157]. In another study performed on a Japanese population, the hazard ratio of cancer per doubling of the probability of T2DM was 0.90 (95% CI = 0.74–1.10) [158].

#### 2.4.7. Kidney Cancer

Kidney cancer is also associated with DM. A meta-analysis showed that the RR of kidney cancer in diabetes is 1.42 (95% CI = 1.06–1.91), however, this association is stronger in women (RR = 1.7; 95% CI = 1.48–1.97) [159]. End-stage renal disease (ESRD), also known as end-stage renal failure, a frequent disease associated with T2DM, is also suggested as a risk factor for kidney malignancy [160]. An association of chronic kidney disease (CKD), also called chronic kidney failure, with the lesser stage of ESRD was also found with renal cancer [160]. Obtained results have shown that the most common cause of CKD and ESRD is T2DM [161]. Higher circulation levels of carcinogens and toxins, as well as the inhibition of immunity due to disturbances in the excretory function of the kidneys in CKD, may be involved in relation to CKD-cancer [160]. Investigation specimens of tumor nephrectomy showed different histopathological changes in non-invaded renal parenchyma. Normal renal tissue adjacent to tumors was detected only in 10% of cases [162]. Pathologic alterations associated with DM, such as glomerular hypertrophy, mesangial expansion, and arteriolar hyalinosis, were the second most frequent changes detected in neoplastic renal tissue samples of tumor nephrectomy [163]. Results obtained from other research revealed that, at the time of kidney cancer diagnosis, 25.4% of these patients had T2DM [164]. Associations between T2DM and kidney cancer were also described by other authors. Weak evidence was found for a positive association between T2DM and kidney cancer [8]. A comprehensive meta-analysis regarding T2DM and the risk of kidney cancer and mortality revealed that T2DM increases the risk developing of kidney cancer RR = 1.38 (95% Cl = 1.10–1.72) [124]. Different associations of the risk of kidney cancer development were detected depending on the type of DM (T1DM or T2DM), as well depending on sex [42].

#### 2.4.8. Urothelial Cancer

Urothelial cancer diagnosis is also associated with DM, and T2DM has been recognized as an independent predictor of worse outcomes after the diagnosis of urothelial cancer [165] and bladder cancer (BC) risk evaluations in DM [166]. BC is one of the most prevalent malignancies in the world [167]. A performed analysis showed an association between T2DM and BC [109]. Meta-analyses of T2DM and the incidence of BC revealed an increased RR = 1.35 (95% CI = 1.17–1.56) [124]. Most studies on the association between T2DM and cancer risk have been conducted on Western populations [17]. A meta-analysis of 36 observational studies on BC showed that most studies were carried out in Western countries, and only one study was performed in Korea [168]. Based on this information, it is postulated that the current results cannot fully represent the worldwide association of T2DM and BC [31]. Moreover, there are different results. For example, a performed meta-analysis showed a negative relationship between BC and the duration of T2DM. Based on this meta-analysis, patients with T2DM for less than 5 years have a higher risk of developing bladder cancer [169]. On the other hand, a case–control study revealed totally different results; the risk of BC increases with the duration of T2DM. The obtained results were as follows: OR = 1.92 for 1–5 years, 1.63 for 5–10 years, 2.39 for 10–15 years, and 2.58 for ≥15 years [170]. A cohort study confirmed a positive relationship between T2DM and BC in women [171]. On the other hand, a meta-analysis showed that the associations between DM and increased risks of BC or BC mortality in women need further explanation [172]. Also, the findings from epidemiological studies are controversial [167,169,173].

#### 2.4.9. Thyroid Cancer

Thyroid cancer incidence has increased greatly in the past three decades [174]. In the endocrine system, this cancer is the most common malignancy. For example, in 2022, 43,800 new cases were diagnosed, with a threefold higher overall incidence rate in American women [175]. In Chinese women, it is the fourth most frequent cancer, with this incidence rising by 12.4% every year [176,177]. Thyroid cancer is also the most common cancer in Korea, especially among women. In 2009, a total of 31,811 new cases of thyroid cancer were diagnosed, with 26,682 cases (73.9%) observed in women [178]. It is postulated that the rapidly increasing incidence of thyroid cancer is predominantly caused by overdiagnosis. It is not clear whether the increased incidence of this cancer is also due to exposure to certain risk factors [179], which are not well established, although it is known that neck irradiation and follicular thyroid cancer are risk factors [180,181]. The roles of other risk factors in the development of thyroid cancer need further investigation [182]. However, as the incidence of thyroid cancer increases, its survival remains high [178]. A possible role of thyroid cancer has been suggested. This suggestion is based on the most rapid increase in T2DM prevalence. A performed meta-analysis revealed that thyroid cancer patients with T2DM had a 1.34-fold higher risk for thyroid cancer when compared to thyroid cancer patients without DM (95% CI = 1.11–1.63) [182]. It was also found that T2DM was associated with a 1.38-fold increased risk in women (95% CI = 1.13–1.67), as compared to nondiabetic women, whereas in men, the risk of thyroid cancer did not remain significant (RR = 1.11; 95% CI = 0.80–1.53). A more detailed analysis of the obtained results revealed that women with pre-existing DM have an increased risk of thyroid cancer [182]. Another meta-analysis of cohort studies revealed that diabetic patients had a 1.32-fold higher risk of thyroid cancer (95% CI = 1.22–1.44) in comparison with thyroid cancer patients without DM. The analysis obtained results depending on sex, showing that these values were 1.26-fold (95% CI = 1.12–1.41) higher in men and 1.36-fold (95% CI = 1.22–1.52) higher in women. A more detailed analysis revealed that the RR of thyroid cancer in patients with DM was 1.34 (95% CI = 1.17–1.53) in the population study. In men, these values were 1.32 (95% CI = 1.12–1.54), in women they were 1.37 (95% CI = 1.12–1.68), and in women with gestational diabetes, the RR of thyroid cancer was 1.30 (95% CI = 1.17–1.43). In women with T1DM, the risk of thyroid cancer was 1.51-fold higher, but not in men [71]. The results presented above confirm the association of DM with thyroid cancer. This association is confirmed in another study [183].

#### 2.4.10. Hematologic Malignancies

Hematologic malignancies seem to be associated with DM. A performed meta-analysis in one study showed a greater risk of non-Hodgkin’s lymphoma, especially peripheral T-cell lymphoma, leukemia, and myeloma, however, not of Hodgkin’s lymphoma among diabetic patients with cancer [184]. Research carried out in New Zealand revealed a 11% increased risk of developing non-Hodgkin’s lymphoma in patients with T2DM and an RR = 1.11 of this cancer in diabetic patients with cancer in comparison with cancer patients without DM. An RR = 1.29 and RR = 1.15 were observed in the case of myeloma and leukemia, respectively. These values were dependent also on sex and age [110]. The results obtained from a prospective study revealed no association between DM and the risk of non-Hodgkin’s lymphoma overall in men (HR = 1.28; 95% CI = 0.89–1.84), in women (HR = 0.71; 95% CI = 041–1.25), and in men and women combined (HR = 1.09; 95% CI = 0.80–1.47). In the case of the B-non-Hodgkin’s lymphoma subtypes, a statistically significant increased risk of B-cell chronic lymphocytic leukemia was observed in men (HR = 2.0; 95% CI = 1.04–3.86), but not in women (HR = 1.07; 95% CI = 0.33–3.43) [185].

#### 2.4.11. Lung Cancer

Lung cancer is the most frequently diagnosed cancer and is associated with cancer-related deaths [3]. The major risk factor for lung cancer is smoking, however, insulin resistance and T2DM also increase the risk of lung cancer in smokers [186]. A large cohort study investigated whether T2DM can be associated with increased a lung cancer incidence among never smokers [187]. The obtained results revealed that T2DM does not increase the risk of lung cancer among never smokers. The hazard ratios and 95% confidence interval (HR = 0.91; 95% CI = 0.71–1.17) suggested no significant association of T2DM with lung cancer incidence. T2DM has a minimal impact on the development of lung cancer in never smokers [187]. Other research investigated the age-standardized rates (ASR) for lung cancer incidence in patients with DM and without DM. The obtained results revealed ASR without diabetes, 85/100,000, and with diabetes, 125/100,000. Considering dependence on sex, these results were: for female patients without diabetes, 76/100,000 and with diabetes, 119/100,000. For male lung cancer ASR without diabetes, 96/100,000, and with diabetes, 132/100,000 [110]. The analysis of results on the association between T2DM and lung cancer incidence revealed: RR =1.03, 95% CI = 0.94–1.13, and the evidence was graded as not significant between T2DM and lung cancer [8]. 

#### 2.4.12. Breast Cancer

Breast cancer is the main carcinoma in women in high-income countries, and in low- and middle-income countries, its incidence shows a tendency for rapid growth [31]. Performed investigations showed an influence of DM on a higher incidence and greater mortality rates of breast cancer [188]. Based on a meta-analysis, it has been suggested that the association between T2DM and breast cancer is confined to post-menopausal women [189]. On the other hand, research has shown an association of breast cancer with DM in pre-menopausal women [190]. More detailed studies confirmed a contribution of DM to an increased risk of breast cancer. Among Asian-American women with diabetes, after adjusting for BMI and waist–hip ratio (WHR), the incidences of breast cancer still increased [191]. There are also other studies which have confirmed the correlation between diabetes mellitus and breast cancer [31]. A meta-analysis revealed a moderate association between breast cancer and T2DM (RR = 1.25; 95% CI = 1.20–1.29) [192]. Therefore, the influence of T2DM on stages at breast cancer diagnosis was examined. A retrospective cohort study examined stage at diagnosis (II, III, or IV vs. I) among women who were newly diagnosed with invasive breast cancer [193]. In this study, women with diabetes were compared to those without DM, and the results showed that diabetic women had significantly more advanced-stage breast cancer in comparison with nondiabetic women with breast cancer. T2DM was associated with an increased risk of Stage II (adjusted odds ratio (aOR) 1.14; 95% CI 1.07–1.22), Stage III (aOR = 1.16; 95% CI = 1.11–1.33), and Stage IV (aOR = 1.16; 95% CI = 1.01–1.33) vs. Stage I breast cancer. It was also found that women with T2DM had a higher risk of lymph node metastases (aOR = 1.16; 95% CI = 1.06–1.27), as well as tumors with size over 2cm (aOR = 1.16; 95% CI = 1.06–1.28). The authors suggested that DM may cause more aggressive breast cancer and may be associated with increased cancer mortality (HR = 1.57; 95% CI = 1.23–2.01) [193]. The age-standardized rate for breast cancer without diabetes was 226/100 000, and with diabetes, 262/100,000 [110]. Highly suggestive evidence of T2DM and great risk of breast cancer incidence (RR = 1.20; 95% CI = 1.12–1.28) was found [8]. An increased risk of breast cancer due to DM was also described in other research [41]. Researchers have found significantly increased risks of breast cancer caused by T2DM (OR = 1.22; 95% CI = 1.07–1.40) and nullified by gestational diabetes (OR = 1.06; 95% CI = 0.79–1.40). In males too, an increased risk of breast cancer was seen, however, the results were not statistically significant (OR = 1.29; 95% CI = 0.99–1.67) [31]. There are also other results. For example, no overall association was found between DM and breast cancer (aHR = 1.02; 95% CI = 0.92–1.14), as well as T2DM not being associated with breast cancer risk overall (aHR = 1.00; 95% CI = 0.90–1.12). However, in the last case, there was a significant risk of breast cancer in the short time after diagnosis of diabetes. Women with T1DM had a higher risk of breast cancer in comparison with nondiabetic women (aHR = 1.52; 95% CI = 1.03–2.23) [194]. In other research, authors observed no overall association between T2DM and breast cancer, however, this risk was increased in the case of triple-negative breast cancer [195]. 

#### 2.4.13. Endometrial Cancer

Endometrial cancer (EC) is the fourth most common cancer in women and is the most common type of gynecological cancer in high-income countries. In the European Union, every year, more than 88,000 new cases are diagnosed. More than 90% of cases are diagnosed in women >50 years of age [196]. EC is closely associated with endometrial hyperplasia and unopposed estrogen exposure [197]. In most epidemiological studies, it has been demonstrated that DM is an important risk factor for endometrial cancer. However, many studies did not distinguish between T1DM and T2DM, but based on the prevalence of these types of diabetes, the vast majority of cases are linked to T2DM [198]. The results obtained from performed studies suggest that T2DM and EC are associated with low physical activity and obesity [199,200]. A cohort study performed in Sweden revealed the incidence of EC among women with diabetes, with an SIR = 1.8; 95% CI = 1.6–2.0. This result confirms that DM increases the risk of EC. A case–control study in Washington showed that T2DM is strongly related to EC (OR = 1.7; 95% CI = 1.2–2.3). These studies also revealed that new-onset DM (<5 years) has 2-fold increased odds of EC as compared with women with a more distant diagnosis (≥5 years) [201]. A meta-analysis performed on cohort studies revealed an increased morbidity of EC in women with diabetes in comparison with nondiabetic women. The obtained results were as follows: RR = 1.89 (95% CI = 1.46–2.45), and the incidence rate was 1.61 (95% CI = 1.51–1.71), confirming the role of T2DM as an independent risk factor for EC [31]. In other studies, the obtained results also showed an association of T2DM and an increased risk of and comorbidities with EC. In these studies, the RR was 2.10 (95% CI = 1.75–2.53) and HR was 1.81 (95% CI = 1.37–2.41) [202]. This meta-analysis revealed a more than two-fold increase in the risk of EC in women with diabetes. The risk of cancer was stronger among case–control studies (RR = 2.22; 95% CI = 1.80–2.74) in comparison with cohort studies (RR = 1.62; 95% CI = 1.21–2.16). It was also found that this risk was slightly lower in studies performed in the USA as compared to studies performed in Europe [198]. Also, in another study, highly suggestive evidence was detected for T2DM and a greater risk of EC incidence (RR = 1.65; 95% CI = 1.50–1.81) [8]. The association of T2DM with the risk of death due to EC is less clear [203]. Based on the results obtained from a prospective study, a significantly increased age-adjusted risk of death caused by EC was observed (1.72; 95% CI = 1.40–2.15) [204]. T2DM also causes poor survival after incident EC, independent from tumor stage or its grade [205]. Another a prospective study showed a multivariable-adjusted RR = 1.33 (95% CI = 1.08–1.95) and an age-adjusted RR = 1.72 (95% CI = 1.40–2.12) [81]. On the other hand, some studies showed that EC and T2DM had no significant association with mortality [203,206]. Therefore, the correlation between T2DM and EC-specific mortality needs additional study [207]. It was found also that T1DM has a statistically significant positive association with EC (RR = 3.15; 95% CI = 1.07–9.29) [198].

#### 2.4.14. Epithelial Ovarian Cancer

Epithelial ovarian cancer (EOC) is the fifth most common cancer and fourth most common cause of death due to cancer. It is predominantly a cancer of post-menopausal women [208]. The exact cause of EOC remains unknown. Many risk factors of ovarian cancer have been suggested, such as early menarche, late menopause, obesity, smoking, and so on [209,210]. Precursor lesions of cancer have not been found. Therefore, it is suggested that EOC develops de novo [211] or that ovarian carcinomas may be caused by the implantation of malignant cells from a tubal carcinoma to the ovaries [212]. Studies concerning the association between T2DM and EOC are limited, and this association is not clear. It was found, based on a prospective study, that the status and duration of T2DM are not significant associated risks of EOC (T2DM status RR = 1.05; 95% CI = 0.75–1.46; T2DM duration < 10 years RR = 1.04; 95% CI = 0.69–1.57; T2DM > 10 years RR = 1.06; 95% CI = 0.63–1.79) [213]. Small-scale epidemiological studies revealed inconsistent results regarding the association between EOC and T2DM. Another meta-analysis showed that women with DM had an increased risk of EOC (RR = 1.17; 95% CI = 1.02–1.33) [214]. Also, weak evidence was found for the positive association between T2DM and EOC (RR = 1.19; 95% = CI 1.06–1.34) [8]. Women with ovarian carcinoma and T2DM have poorer survival chances in comparison with nondiabetic women with EOC [215].

#### 2.4.15. Cervical Cancer

Cervical cancer is the fourth most frequent cancer in women. A significant association exists between the incidence of cervical cancer and T2DM, and the influence of T2DM on the prognoses for women with cervical cancer still has not been determined [210]. Obtained results suggest that T2DM may increase the risk of cancer recurrence and death for women with early-stage cervical cancer, also after curative treatment [216]. Early-stage cervical cancer in women with T2DM has a poorer oncological outcome than nondiabetic women [217]. The relative difference in the rates between women with diabetes in comparison to nondiabetic women was the highest for uterine cancer (RR = 2.98; 95% CI = 2.77–3.19]. An analysis of these female-specific cancers revealed that the strongest relative difference in the incidence of cancer between female patients with diabetes in comparison to those without T2DM is in the case of uterine cancer [110].

#### 2.4.16. Prostate Cancer

Prostate cancer is the second leading cause of cancer death in American males and newly diagnosed cases of this cancer are also highest in the USA [218]. Performed studies revealed an inverse association between T2DM and prostate cancer [219]. Also, several other studies confirmed the association of DM with a reduced risk of prostate cancer [220]. Another study revealed that the rate of prostate cancer was 15% lower among patients with diabetes compared to patients with prostate cancer without diabetes (RR = 0.85; 95% CI = 0.82–0.88), ASR without DM, 321/100,000, and with DM, 273/100,000 [110]. It is worth noting that the inverse association between prostate cancer and T2DM is limited only to incidence, but not mortality. Patients with prostate cancer have a worse prognosis [221]. One Swedish peer study revealed a lack of association between T2DM and prostate cancer [222]. Therefore, this association needs further investigation [31]. 

#### 2.4.17. Testicular Cancer

No difference was found in the rate of testicular cancer between diabetic patients compared with nondiabetic individuals (RR = 0.90; 95% CI = 0.66–1.22) [110].

#### 2.4.18. Melanoma

An inverse association between T2DM and cancer was observed in the case of melanoma. The obtained results for melanoma in women showed ASR without diabetes of 83/100,000 and with diabetes of 65/100,000. For male patients, the obtained results showed ASR without diabetes, 115/100,000, and with diabetes, 93/100,000. The overall trends of melanoma revealed an RR of 0.81 [110].

### 2.5. Mortality of Patients with T2DM and Cancer

The mortality of patients with cancer depends on several factors, such as older age, obesity, physical inactivity, DM, sex, etc. For example, there are different results for diabetic patients with cancer in comparison to nondiabetic patients with cancer. The results may be different for women in comparison with men. The obtained results depend also on the type of DM: T1DM or T2DM. The mortality of patients also depends on the type of cancer (Table 1). 

### 2.6. Association of Specific Cancer with T2DM

Performed meta-analyses and other investigations revealed that, in some cases, the risk of the cancer type is strongly associated with T2DM [36], as presented in Table 2.

## 3. Underlying Mechanisms of Associations between Diabetes Mellitus and Cancer

Diabetes mellitus is associated with several metabolic disturbances and pathologies, such as obesity, insulin resistance, hyperinsulinemia, hyperglycemia, impaired intracellular insulin/IGFs signaling pathway, chronic inflammation, and so on. Genetic factors may also influence DM. The mentioned disturbances and pathologies may be a cause, as well as being an effect of DM. On the other hand, these pathologies may be associated with cancer, causing its initiation, development, progression, and metastasis. Several antidiabetic drugs may stimulate the development of cancer (Figure 1).

### 3.1. Insulin Resistance and Hyperinsulinemia

Insulin resistance and hyperinsulinemia are commonly diagnosed in patients with T2DM. The obtained results suggest that these pathologies are involved in the pathogenesis of diabetes-associated cancer [223,224]. In patients with T2DM, the efficiency of insulin uptake by cells and the utilization of glucose are reduced, causing prolonged hyperglycemia. A compensatory mechanism to obtain normoglycemia is the secretion by pancreatic β-cells of large amounts of insulin, causing hyperinsulinemia. Hyperinsulinemia may be observed also in patients with T1DM due to the administration of exogenous insulin. Many tumors, such as breast, colon, lung, prostate, ovary, and thyroid cancers, overexpress the insulin receptor (INSR) [53]. Elevated levels of circulating insulin and higher levels of insulin receptors are important risks for developing several cancers in diabetic patients. Insulin binding to INSR causes the activation of the tyrosine kinase activity of INSR. The activation of INSR causes the phosphorylation of insulin receptor substrate (IRS), resulting in the activation of phosphatidylinositol 3 kinase and the protein kinase B (PI3K/AKT) signaling pathway. This signaling is associated with most metabolic and mitogenic effects of insulin [225]. The activated PI3K/AKT signaling pathway causes phosphorylation and subsequently activates the mammalian target of rapamycin (mTOR), which activates its downstream signaling pathway. Activating this signaling pathway is involved in the regulation of several processes of cancer such as the survival of cancer cells, proliferation, invasion, migration, differentiation, angiogenesis, and metastasis [223,224,225]. The activation of the PI3K/AKT signaling pathway also stimulates the translocation of β-catenin into the nucleus, causing increased levels of vascular endothelial growth factor (VEGF), which is involved in cancer cell behaviors [226,227]. Hyperinsulinemia due to DM stimulates the secretion of IGF-1 by the liver, as well as elevated insulin levels, upsurging the insulin-like growth factor-1 receptors (IGF-1R). In patients with hyperinsulinemia, IGF-binding protein-1 (IGFBP-1) is inhibited, resulting in an increased biological activity of IGF-1, promoting tumor growth, survival, motility, and drug resistance [14,228]. Also, insulin-like growth factor-2 (IGF-2) may activate the PI3K/AKT/mTOR signaling pathway, stimulating the development of cancer [53]. For more details, see [14,53,111,183,223,224].

### 3.2. Hyperglycemia

The main source of energy in cells is the tricarboxylic acid cycle, also known as the Krebs cycle, a process taking place in mitochondrial oxidative phosphorylation. Cancer cells shift to glycolysis even in the presence of oxygen. This phenomenon is known as the Warburg effect [229]. Therefore, cancer cells require an increased uptake of glucose to obtain sufficient energy. Hence, hyperglycemia, a characteristic state in diabetic patients, provides cancer cells with great conditions for survival and proliferation, etc. The glucose metabolism is also involved in the synthesis of protein and DNA in cancer cells. Therefore, higher blood glucose levels influence cancer growth and metastasis [31]. Increased aerobic glycolysis in cancer cells supplies the substrates for the synthesis of nucleotides, amino acids, and lipids, stimulating the proliferation of cancer cells [224]. There are three rate-limiting enzymes: hexokinase, phosphofructokinase, and pyruvate kinase. The regulation of these enzymes in cancer cells is altered by oncogenes, the loss of tumor suppressors, and the activation of the PI3K/AKT signaling pathway [229]. The increased aerobic glycolysis observed in many cancer cells is due to the upregulation of glycolytic genes and oncogenes, as well as the mutation of tumor suppressor genes [229]. High blood glucose levels trigger several mechanisms involved in the proliferation of cancer cells and their migration, invasion, and immunological escape. Changes in the metabolism of cancer cells due to high-energy utilization or glucose toxicity stimulate the synthesis of reactive oxygen species (ROS) and damage of DNA, resulting in oxidative stress and the inflammatory response. Prolonged elevated blood glucose levels stimulate the production of advanced glycation end products (AGEs), which interact with their specific receptor, activate the nuclear factor kappa-light-chain enhancer of activated B cells (NF-κB), and generate ROS, increasing oxidative stress and causing increased pro-inflammatory signaling [230]. For more details, see [50,111,231].

### 3.3. Obesity

Most patients with T2DM are overweight or obese [54]. Body mass index (BMI) is generally used as a measure of obesity. The fat distribution in the body is also important, so, therefore, as a measure of obesity, waist circumference is preferred over BMI. There are many results obtained in several studies suggesting that waist circumference, waist-to-hip ratio, and direct measures of visceral adiposity are associated with a risk of cancer development, independently of BMI [232]. Abdominal obesity, also known as visceral, central, upper-body, or android obesity is more dangerous than gynoid obesity, because it increases the risk of cancer. Waist circumference correlates better with visceral obesity. Abdominal obesity is the type that may be most involved in the development of insulin resistance, dyslipidemia, glucose intolerance, and hypertension, all disturbances associated with the development of T2DM [53]. This type of obesity may stimulate metabolic abnormalities, resulting in the impaired release and signaling of several hormones, adipokines, inflammatory cytokines, growth factors, and free fatty acids, as well as increased levels of pro-inflammatory mediators [233]. The content of visceral fat is involved in the induction of insulin resistance and hyperinsulinemia [234]. 

A large peer study including about 1 million adults from the USA examined the association between BMI and cancer mortality. Most obese men and women had a 40–80% increased risk of death caused by cancer [235]. The same group subsequently analyzed and published a more detailed analysis of these results. The obtained results revealed that severe obesity was associated with a significantly increased overall cancer mortality. A linear trend of increased mortality with an increasing BMI and significantly increased mortality from colorectal, liver, gallbladder, pancreatic, esophageal, and kidney cancer, non-Hodgkin’s lymphoma, and multiple myeloma was found in both sexes. In women, an increased risk of death was observed in the case of breast, endometrial, cervical, and ovarian cancer, while in men, this was in cases of leukemia, esophageal, stomach, and prostate cancer. Also, an inverse association was found between BMI and lung cancer mortality in both sexes, whereas no association was observed between melanoma, brain, and bladder cancer in both sexes [236]. A lower incidence of cancers related to obesity [237] and a significant decrease in medical care associated with cancer [238] were observed in patients after bariatric surgery, in comparison with morbidly obese individuals. 

Obesity may play different roles in the relationship between T2DM and cancer [124]. The associations of T2DM with increased risks of cancer are probably due to mechanisms that contribute to cellular proliferation, inflammation, and hormonal balance [239]. These mechanisms are also suggested in the association between T2DM and cancer. The influence of obesity and T2DM on pancreatic ductal pathology was studied, and the results obtained revealed that the epithelial replication of the pancreatic duct was increased ten-fold in samples obtained from obese patients without T2DM in comparison with lean individuals without diabetes. It was found also that ductal epithelial replication was increased four-fold in lean patients with T2DM as compared to lean patients without diabetes. Based on the obtained results, the researchers suggested that T2DM and obesity are independent factors for the development of pancreatic exocrine neoplasia [240].

As mentioned above, obesity may change the metabolism of adipose tissue, resulting in the increased secretion of hormones, adipokines, inflammatory cytokines, growth factors, etc. [233,241]. These adipose-tissue-specific secreted compounds may cause the initiation and progression of several cancers due to the metabolic reprograming of cells [242,243]. To date, more than 600 adipocyte-enriched secretory factors have been described. Adipokines, bioactive polypeptides, play an important role in the regulation of several processes, such as the homeostasis of glucose and energy, which are also involved in metabolic pathways [244]. The increased amount of adipose tissue in obese people alters the secretion of adipokines, causing chronic low-grade inflammation, resulting in metabolic disturbances. The dysregulated secretion of adipokines influences the cellular physiology of tumor cells, contributing to the growth of cancer cells and their proliferation [245], migration, and invasion [246], epithelial–mesenchymal metastasis [247], and the development of multidrug resistance [248]. Adiponectin and leptin are two of the most studied adipokines which are associated with cancer development. Adiponectin protects against development and progression of cancer associated with obesity [249]. Low levels of adiponectin are associated with metabolic disorders, including T2DM, obesity, and the development of cancer [250]. Low levels of adiponectin are associated with an increased cancer risk, whereas higher levels of adiponectin are associated with a decreased cancer risk and a reduction in cancer progression [251]. Obtained results revealed that adiponectin regulates different intracellular signaling pathways associated with cancer. Upon binding to its receptor, AdipoR1 and AdipoR2, it activates signaling pathways such as monophosphate-activated protein kinase (AMPK), mTOR, PI3K/protein kinase B (AKT), MAPK, peroxisome proliferator-activated receptor γ (PPARγ), signal transducer and activator of transcription 3 (STAT3), and nuclear factor-κB (NF-κB) [14]. Increased concentrations of plasma leptin are associated with obesity. It was found that elevated levels of leptin and the overexpression of its receptor (Ob-R) stimulate the intracellular signaling pathways associated with cell proliferation, migration, invasion, metastasis, and the epithelial–mesenchymal transition in breast cancer [252]. Within tumor tissue, leptin enhances different responses of tumor cells due to the aberrant activation of intracellular signaling pathways, such as MAPK kinase (MEK)/extracellular signal-regulated kinase (ERK)1/2, Jak/STAT3, and PI3K/AKT [252].

The cancer microenvironment plays a key role in tumorigenesis [253]. The extracellular matrix (ECM) and hypoxia regulate the tumor microenvironment. The remodeling of the ECM due to cancer changes different properties of the tumor microenvironment, creating a favorable microenvironment for tumor growth, migration, and metastasis [254]. Rapidly proliferating cancer cells have a high oxygen demand, resulting in an insufficient supply of oxygen to the cancer, which becomes hypoxic [255]. Hypoxia stimulates cancer angiogenesis, metastasis, and resistance to therapy [256] and inhibits anti-tumor immune cells [257,258]. Obesity influences the cancer microenvironment due to dysfunctional adipose tissue and impaired extracellular signals, resulting in the stimulation of cancer growth, proliferation, invasion, migration, and metastasis, as observed in breast cancer [259]. Adipose tissue hypoxia causes a proinflammatory microenvironment that is associated with a microenvironment for cancer promotion [253]. An impaired function of adipose tissue changes the production of adipokine, especially releasing proinflammatory cytokines [243]. This imbalance of adipokine contributes to a favorable microenvironment for cancer growth and progression [14]. Obesity, increasing oxidative stress [260], may contribute to cancer progression and metastasis [261].

### 3.4. Age

The development of some cancers is detected in childhood and young adults. But the incidence of most cancers increases with age. For example, 78% of all newly diagnosed cancers are detected in individuals aged ≥55 years [262]. The incidence of diabetes is also associated with age. For example, in American adults, the prevalence of diabetes is 2.6% in individuals aged 20–39, in people aged 40–59 years, the prevalence of disease is 10.8%, and in men ≥60 years, the number of diabetic patients increases to 23.8% [100,232]. Advanced age is one of the important factors which may cause an increased risk of insulin resistance, which is a common symptom associated with T2DM. Increased age decreases the secretion of insulin and tolerance to glucose [263]. There are also other factors associated with an increased risk of insulin resistance in older people. An example may be free radicals, which induce oxidative stress and disturb the function of the mitochondria [264], as well as decrease the insulin-stimulated metabolism of glucose in the muscles, increased accumulation of fat in the liver and muscles, and a 40% reduction in mitochondrial oxidative phosphorylation. These pathologies may be due to insulin resistance, which, as mentioned above, is a common compound of T2DM [265]. 

### 3.5. Inflammatory Cytokines

Inflammation is a strong link between DM and cancer [266]. Chronic inflammation characterized by high levels of oxidative stress, ROS, the activation of pro-inflammatory pathways, and the impaired production of adipokine, may stimulate cancer cell growth, enhance metastasis, increase angiogenesis, and disturb the function of natural killer cells and macrophages [267]. The production of inflammatory cytokines is associated with obesity, especially with visceral adiposity, a characteristic feature observed in DM. Adipose tissue produces free fatty acids, interleukin-6 (IL-6), monocyte chemoattractant protein, plasminogen activator inhibitor-1 (PAI-1), adiponectin, leptin, and tumor necrosis factor-α (TNF-α). These factors may be involved in the regulation of the malignant transformation or cancer progression. The activation of signal transducer and activator of transcription protein (STAT) signaling, with the association of cytokines such as IL-6, enhances the proliferation, survival, and invasion of cancer cells, whereas host-anti-tumor immunity is suppressed [268]. TNF-α and IL-6 are the major inflammatory cytokines associated with diabetes and cancer. IL-6 activates NF-κB and increases the levels of cyclin D1, causing the development of neoplastic transformation. This cytokine causes cells to isolate from each other. With the activation of the process of the epithelial-to-mesenchymal transition, these cells remain alive, resulting in cancer metastasis [269]. TNF-α in nondiabetic individuals plays a role as an important mediator of anti-tumor immune responses. Chronic exposure to TNF-α in diabetic and/or obese individuals activates a series of signaling pathways, such as NF-κB, mitogen activated protein kinase (MAPK), and Jun kinase, and inhibits the apoptosis of cancer cells, stimulating growth and metastasis of these cells [260].

### 3.6. Advanced Glycation End Products

Reactive carbonyl species (RCSs) are unstable carbonyl compounds, especially aldehydes, as a result of the oxidative and non-oxidative reactions of carbohydrates and lipids [270]. In diabetes and metabolic disorders due to obesity, increased levels of RCS may derive from glucose [271], which are increased in T1DM and T2DM as an effect of hyperglycemia, or from lipids [272]. RCSs may react with free amino groups and thiol groups, causing physico-chemical modifications to biomolecules, such as proteins, lipids, and nucleic acids, affecting many functions of these molecules. Depending on the substrate from which the RCS originates, the final products of the reaction are defined as advanced end products (AGEs) in the case of carbohydrates, or advanced lipoxidation end products in the case of lipids [43]. A higher availability of glucose, caused by hyperglycemia and/or lipids as an effect of dyslipidemia, is the main mechanism that increases carbonyl stress in diabetes and other metabolic disturbances. Long-term increased blood glucose levels, hyperglycemia, accelerate the formation of AGEs. During the ageing process, AGEs accumulate in sera and tissues due to oxidative and glycolytic reactions and reduced activity of the detoxification system [273]. The rate of this build-up is accelerated in several disease conditions, such as obesity, diabetes, as well as others [270]. There are also exogenous AGEs, for example, derived from the diet, which contribute to several diseases, including cancers [274]. The binding of AGEs to their receptors (RAGEs) causes the activation of transcription factors and redox-sensitive signaling pathways, resulting in the formation of reactive oxygen species (ROS), inflammation, proliferation, and so on [275]. Therefore, AGEs are associated with the development of several diseases, such as diabetes and cancers. Carbonyl stress can affect the homeostasis of cell and tissue, in a contribution with several mediators, such as RCS, the AGE/RAGE axis, and ROS. Each of these mediators are able to affect the cellular redox status, causing the activation of the redox-sensitive NF-κB, which regulates hundreds of genes associated with cellular stress and survival. It is suggested that therapeutic strategies against carbonyl stress should be associated with a reduction in AGE formation, for example, by the enhancement of RCS degradation, inducing of detoxifying enzymes. A blockade of RAGE signaling and the use of RVS scavengers, such as derivatives of hydrazine, vitamin B, amino acids, [276] and several others, have been suggested. It is also proposed to use polyphenolic compounds from plants as therapeutic factor, which may inhibit glycation and the formation of AGEs. There are some polyphenolic compounds which regulate the blood glucose metabolism due to the amplification of cell insulin resistance and activation of an insulin-like growth factor binding protein signaling nature. These compounds show also antioxidant properties and metal chelating activity and, therefore, they could be a possible mechanism against glycation and the formation of AGEs [277]. 

### 3.7. Other Mechanisms Involved in Associations between Diabetes Mellitus and Cancer

The above-mentioned factors are the main and common factors involved in the association between diabetes mellitus and cancer. There are also several other factors that may play roles as links between diabetes mellitus and developing cancer. Examples of these factors include genetic factors, especially mutations in the insulin/IGFs signaling pathway, dyslipidemia, hormones, including sex hormones, and anti-diabetic drugs such as insulin and insulin analogs [14,30,31,50,53,109]. On the other hand, a healthy lifestyle may cause decreased incidences of DM and cancer [278,279].

## 4. Anticancer Therapies Which May Cause Hyperglycemia and Insulin Resistance

There are several anticancer drugs and therapies which may cause hyperglycemia and/or insulin resistance. Over the past two decades, the anticancer therapy for most solid and hematological malignancies has changed. In the case of conventional cytotoxic agents, there is a minimal or no direct effect on hyperglycemia observed. Unfortunately, novel drugs and immunotherapy may induce hyperglycemia and insulin resistance [24]. The rate of hyperglycemia depends on the cancer type, as well as on other factors, such as comorbid illness and other medications used, etc. There several side effects that have been observed. Some immunotherapies in which PD-1 (programmed cell death-1) inhibitors, PDL-1 (programmed cell death ligand-1) inhibitors, or cytotoxic T lymphocyte-associated antigen-4 (CTLA-4) are used may cause the decreased production of insulin in pancreatic islet β-cells, as has been observed in patients with T1DM. Inhibitors of the PI3K/mTOR signaling pathway may cause insulin resistance, as in the case of T2DM.

### 4.1. Immunotherapy

Immunotherapies modulating the immune response against cancer cells play an important role in cases of metastatic melanoma, non-small-cell lung cancer, renal cell carcinoma, bladder carcinoma, and head and neck cancer and lymphomas. Anti-CTLA-4 (ipilimumab), PD-1 (nivolumab, pembrolizumab), and PDL-1 inhibitors (atezolizumab, avelumab) trigger an immune response, activating T cells and the immune response against cancer cells [9,24].

### 4.2. Targeted Therapy

Targeted therapy may affect aberrant signaling pathways and proteins in cancer cells, inhibiting cell proliferation and inducing apoptosis. Unfortunately, as mentioned above, some inhibitors of the PI3K/mTOR pathway cause hyperglycemia and induce insulin resistance.

#### 4.2.1. Inhibitors of PI3K/mTOR Pathway

This pathway is commonly activated in different cancers [280]. Its inhibition may induce insulin resistance. Inhibitors of PI3K, such as alpelisib, have efficacy in women with advanced hormone-receptor-positive breast cancer with *PIK3CA* mutations, but in combination with fulvestrant and endocrine therapy [281]. However, women who received alpelisib developed different rates of hyperglycemia [281].

Inhibitors of AKT, such as the selective inhibitors ipatasertib and capivasertib, pan-AKT kinase inhibitors, cause hyperglycemia in 13–45% of patients with different solid tumors [24]. AKT is protein kinase frequently activated in human cancers.

Inhibitors of mTOR, such as evordimus, sirolimus, and temsirolimus, are used in anticancer therapy. Evorolimus, an orally administered analog of rapamycin, is approved for the management of advanced breast cancer, renal cancer, and pancreatic neuroendocrine tumors [24]. In patients with the tumors mentioned, hyperglycemia is observed in 10–50% of individuals. Temisirolimus is a highly specific inhibitor of the mTOR pathway, administered intramuscularly. It is used in patients with advanced solid malignancies, such as renal cell cancer. In patients who received this inhibitor, hyperglycemia was detected in about 9% of cases [282]. Hyperglycemia associated with use of mTOR inhibitors is likely caused by a combination of impaired insulin secretion and insulin resistance [283]. 

#### 4.2.2. Inhibitors of IGF-1R

Insulin-like growth factor-1 is a cancer-promoting factor. Binding to its receptor, it activates the Ras/MAPK/ERK and a second pathway, the PI3K/AKT/mTOR pathway [284], causing cell proliferation and resistance to apoptosis [285]. Inhibitors of IGF-1, such as monoclonal antibodies and small molecules, commonly induce hyperglycemia. Hyperglycemia was detected in a few patients enrolled in studies with the anti-IGF-1R antibody [286,287]. It may be an effect of a compensatory increase in the concentration of growth hormones (GH) caused by the blockage of IGF-1 with insulin resistance [288].

There are monoclonal antibodies against IGF-1RF, such as dalotuzumab, which is a specific monoclonal antibody against IGF-1R and does not react with the insulin receptor [289]. Its use, in combination with other drugs such as irinotecan and cetuximab, in patients with metastatic colorectal cancer did not improve survival, however, it induced high hyperglycemia (21%) as compared to patients who received placebo (5.2%) [290]. 

The other inhibitors are small-molecule inhibitors of IGF-1R and INSR. An example of this inhibitor may be linsitinib, an oral dual inhibitor of IGF-1R and INSR. The influence of this anticancer drug on the development of hyperglycemia was investigated in patients with advanced adrenocortical tumors [291]. The obtained results revealed that it did not improve survival compared with placebo. The incidence of hyperglycemia associated with linsitinib was 2% higher in comparison with the monoclonal antibody.

There are also other inhibitors of IGF-1R. For example, ganetespib, which is a heat shock protein 90 inhibitor. It blocks several oncogenic pathways, such as IGF-1R, epidermal growth factor receptor (EGFR), VEGF, c-MET, a transmembrane tyrosine kinase that binds to the hepatic growth factor (HGF) and is involved in the ability to metastasize, and human epidermal growth factor 2 (HER2). Its use in patients with hepatocellular carcinoma revealed hyperglycemia in 21% of patients [292]. On the other hand, in patients with advanced lung cancer, ganetespib-related hyperglycemia was not reported [293]. Another IGF-1R inhibitor, certinib, is oral tyrosine kinase inhibitor of anaplastic lymphoma kinase (ALK), however, it is a less potent inhibitor of IGF-1R. In patients with lung cancer treated with certinib, any grade of hyperglycemia was detected in 10% [294].

To note, currently, inhibitors of IGF-1R are not in use in clinical practice [24].

#### 4.2.3. Inhibitors of EGFR

EGFR belongs to the family of HER2/neu (ErbB2), HER3 (ErbB3), and HER4 (ErbB4). Inhibitors of tyrosine kinase EGFR may be used as monoclonal antibodies (cetuxinib and panitumumab) or as small molecules (geftinib, erlotinib, afatninib, and osimertinib). Monoclonal antibodies are commonly used in patients with gastrointestinal cancers, whereas small molecules are used in the case of EGFR-mutated lung cancer [295,296]. Both of the above-mentioned groups of EGFR inhibitors do not cause hyperglycemia, because the pathway of EGFR is not directly associated with glucose metabolism. But, a third-generation inhibitor of EGFR, rociletinib, which is used in patients with EGFR-mutant non-small cell lung cancer with the T790M resistance mutation [297], may induce hyperglycemia in about 22% of patients [24]. 

#### 4.2.4. Inhibitors of BCR-ABL Multi-Targeted Tyrosine-Kinase

Examples of these inhibitors may be nilotinib, dasatinib, and ponatinib, which are multi-targeted tyrosine-kinase inhibitors. These inhibitors are used in patients with chronic myelogenous leukemia (CML) and Philadelphia chromosome-positive acute lymphoblastic leukemia [298]. However, these inhibitors are more effective than imatinib [298], and, unfortunately, they are associated with a high risk of hyperglycemia, which was observed in 36% of patients with grade 3 hyperglycemia and 6% of patients with grade 4 hyperglycemia treated with 300 mg of nilotinib, and 41% of patients (grade 3 hyperglycemia) and 4% (grade 4 hyperglycemia) in the case of 40 mg nilotinib. In the case of imatinib, these values were 20% (grade 3 hyperglycemia) and 0% (grade 4 hyperglycemia) [299]. It is suggested that hyperglycemia associated with these inhibitors is due to tissue insulin resistance and compensatory hyperinsulinemia [300].

#### 4.2.5. Glucocorticoids

Glucocorticoids are commonly used in patients with cancers of the blood system [301], such as lymphomas, leukemias, and myeloma, as well as in the case of solid cancers such as prostate cancers, and to reduce edema in patients with metastasis to the brain and spinal cord [24]. These drugs are also used to treat cancer pain, side effects caused by chemotherapy, such as nausea and vomiting, and cancer-related cachexia [302,303]. Corticoids induce hyperglycemia [304,305]. It is suggested that hyperglycemia or new-onset diabetes mellitus after the administration of glucocorticoids may be due to impairing pancreatic β-cells’ functions and insulin sensitivity [22].

#### 4.2.6. Hormonal Therapies

The first examples of targeted therapy were hormonal therapies targeting estrogen and progesterone receptor signaling. The results obtained from anti-estrogen therapy revealed that, in menopausal women with breast cancer, a high incidence of insulin resistance and diabetes mellitus [306] was observed. Tamoxifen increased by 2.2-fold the risk of diabetes and inhibitors, and aromatase increased by 4.2-fold the risk of diabetes [307].

In the management of prostate cancer, androgen deprivation therapy is commonly used, with analogs of gonadotropin-releasing hormone or antagonists with or without oral antiandrogen. Therapy with these drugs is associated with increased insulin resistance and the incidence of diabetes mellitus [306].

Somatostatin analogs, such as ocreotide and lanreotide, are used in patients with neuroendocrine tumors such as secretory insulinomas, VIPomas, glucagonomas, etc. [308,309,310]. In 2% to 27% of cases, all-grade hyperinsulinemia was observed [24].

#### 4.2.7. Other Compounds

There are also other miscellaneous compounds. In the management of acute lymphoblastic leukemia, asparaginase is commonly used. In pediatric cancer patients, hyperglycemia is noted in 2.5% to 23% of cases [311]. In the management of insulinoma, diazoxide, an activator of the potassium channel, is used, which blocks the secretion of insulin in pancreatic β-cells, inducing hyperglycemia [312].

#### 4.2.8. Chemotherapy

Classical chemotherapies, employing cytotoxic antibiotics, alkylating agents, anti-microtubules agents, antimetabolites, and inhibitors of topoisomerase, are used in the treatment of patients with cancer. Observations performed in vitro revealed that the efficacy of different chemotherapeutics depends on the glucose concentrations. For example, in breast cancer cell lines, doxorubicin, paclitaxel, and 5-fluorouracil (5-FU) induced a cell death, but high glucose concentrations decrease the effects of these drugs [313]. In triple-negative breast cancer lines, high glucose concentrations increased the sensitivity of paclitaxel [314]. There is a problem with investigating the effects of chemotherapeutics on the whole-body metabolism due to the conjunction of chemotherapeutic drugs with glucocorticoids. As mentioned earlier, glucocorticoids induce insulin resistance, also causing hyperinsulinemia. For example, in patients with colorectal cancer treated with 5-FU, 23% developed hyperglycemia, of which 11% had impaired fasting glucose and 12% diabetes [315].

The metabolic consequences of chemotherapeutic drugs treatment have not been widely studied. The obtained results suggest that these drugs may induce insulin resistance and hyperinsulinemia in cancer patients [316].

## 5. Hypoglycemic Therapies Which May Be Associated with Cancer Risk

As mentioned earlier, there are anticancer therapies which may stimulate DM. On the other hand, there are antidiabetic drugs which may be associated with cancer development.

### 5.1. Biguanide Derivatives

Metformin, a biguanide derivative, is nearly 100 years old [84]. It has been widely used as a first-line treatment for hyperglycemia and T2DM for over 50 years [111]. It is used also in therapy for patients with diabetic nephropathy, polycystic ovary syndrome, and cardiovascular complications associated with T2DM and GDM [317]. It is an insulin sensitizer. Primarily, this biguanide decreases insulin resistance in the peripheral tissues, increasing the uptake of glucose, reducing liver glycogen output, inhibiting hepatic gluconeogenesis, and reducing the intestinal absorption of glucose, resulting in decreased blood glucose levels [317]. The first case–control study in 2005 revealed that metformin decreases the risk of developing cancer in diabetic patients [318]. Further, other observational studies confirmed the anticancer effect of metformin [319,320]. The usage of metformin decreases the incidence of patients with different types of diabetes [321] and decreases the mortality in diabetic patients [322]. Observations performed in Taiwan showed that diabetic patients treated with metformin for ≥3 years has a reduced risk of colon cancer (HR = 0.646; 95% CI = 0.490–0.852) [323]. Other research showed an overall reduction of approximately 50% in gastric cancer in users treated with metformin as compared to non-users of metformin [39,324]. In patients with thyroid cancer treated with metformin, a reduced tumor volume was observed. The authors also suggested that a lack of metformin treatment has a negative effect on the complete remission rate and progression-free survival in thyroid cancer [325,326]. It was also found that the effect of metformin on thyroid cancer is time- and concentration-dependent, causing an overall significantly-decreased risk of thyroid cancer [327]. Based on the obtained results, the researchers suggested that metformin should be preferred as a therapy in diabetic patients with thyroid cancer [185]. Metformin decreases the risk of death from pancreatic cancer (HR = 0.79; 95% CI = 0.70–0.92) [328], colorectal-cancer-specific mortality (HR = 0.66; 95% CI = 0.50–0.87) [329], and all-cause mortality in the setting of breast cancer (RR = 0.652; 95% CI = 0.488–0.873); however, no reduction in the incidence of breast cancer in diabetic patients [330] was observed. In patients with endometrial cancer, the usage of metformin caused a higher overall survival rate (HR = 0.82; 95% CI = 0.70–0.95) [331]. The protective role of metformin against risk of cancer was also described in the case of cancers such as estrogen receptor (ER)-positive breast cancer [197], prostate cancer [13], bladder, ovarian, cervical, kidney, oral, esophageal, lung nasopharyngeal, skin, hepatocellular, and biliary tract cancers, and in the case of non-Hodgkin’s lymphoma [39]. A population-based peer study in diabetic patients indicated that metformin therapy is not significantly considered with a decreased risk of cancer in these patients [332,333]. The obtained results also suggested that metformin with T2DM increases the risk of ER-negative breast cancer (HR = 1.25; 95% CI = 0.84–1.88) and triple-negative breast cancer (HR = 1.25; 95% CI = 1.06–2.83) [197]. 

The role of metformin as an anticancer agent has been extensively studied and several potential mechanisms have been suggested. Metformin may decrease cancer incidence, reducing the circulating levels of insulin and IGF-1 in the peripheral blood, decreasing hyperinsulinemia, and improving insulin resistance [53]. Note that insulin and IGF-1 are involved in carcinogenesis due to the upregulation of insulin and the IGF-1 signaling pathway [334]. Observations performed in vitro and in vivo showed that hyperinsulinemia may stimulate cell proliferation, resulting in cancer promotion and progress [335]. Metformin, reducing insulin/IGF-1 blood levels, inhibits the insulin/IGF-1 signaling pathway and improves the cellular metabolism in normal and cancer cells. One major effect of metformin may be the suppression of the mTOR signaling pathway. Reductions in the circulating levels of insulin and IGF-1 in the peripheral blood and the activation of the liver kinase B1 (LKB1)/AMPK signaling pathway cause the inhibition of the mTOR pathway, resulting in cell proliferation, protein translation, and insulin levels [336]. The inhibition of the mTOR signaling pathway reduces the expression of the proto-oncogenes Cyclin D1 and c-MYC, causing the inhibition of the proliferation, migration, and epithelial–mesenchymal transition of thyroid cancer cells [337,338,339]. Other mechanisms of the anti-cancer activity of metformin suggest an inhibition of complex I of the mitochondrial electron transport chain, causing the attenuation of oxidative respiration, resulting in ATP/AMP balance. This disturbance, in turn, activates the LKB1 and AMPK signaling pathway, comprising the inhibition of mTOR, Cyclin D1, and p53 interference, causing antiproliferative effects of the discussed drug [36]. Other mechanisms are also suggested [111,321].

### 5.2. Thiazolidinediones

Thiazolidinediones (TZDs), also called glitozones, are insulin-sensitizing peroxisome proliferator-activated receptor (PPAR)-γ agonists. TZDs improve glucose and lipid metabolism due to an increase in the sensitivity of target cells to insulin [340], reducing insulin resistance [341]. PPAR-γ is a nuclear receptor which with retinoid X receptor (RXR) forms a heterodimer and regulates the expression of genes associated with insulin action, the differentiation of adipocytes, inflammation, and metabolism [342]. The activation of PPAR-γ by TZDs initiates the differentiation of adipocytes, decreases the release of free fatty acids from adipocytes, and the synthesis and release of prostaglandins, TNF-α, interleukin-6 (IL-6), leptin, and resistin by adipocytes. An effect of PPAR-γ is also an increased production of adiponectin, glucose disposal by adipocytes, and insulin sensitivity [342]. However, the main role of TZDs is their metabolic action in T2DM [341], and it is suggested that these drugs may be involved as anti-tumor agents, affecting the cell cycle, apoptosis, and cell proliferation [343]. On the other hand, many clinical studies have showed that treatment with TZDs did not have a significant anti-cancer effect in some cancers [344,345,346]. In 2005, the positive anti-cancer effect of pioglitazone used in patients with T2DM [347] was first suggested. The obtained results revealed that the activator of PPAR-γ induces cell cycle G2 arrest and the inhibition of bladder cancer cell proliferation due to the inhibition of the PI3K/AKT signaling pathway [348]. Observations performed on human cancer cell lines revealed that the induction of apoptosis due to the activation of PPAR-γ inhibits the growth of tumor cells, such as colon, breast, and lung cancer cell lines [349,350]. Pioglitazone reduces proliferative and invasive abilities, induces the apoptosis of non-small cell lung cancer (NSCLC) cells, inhibiting the MAPK/AKT signaling pathway and transforming growth factor β (TGFβ/SMADs), and decreases the risk of developing lung cancer in diabetic patients [351] and colorectal and breast cancer [352,353]. Other studies also revealed that rosiglitazone decreases the risk of thyroid cancer in diabetic patients, particularly those older than 50 years [354], and inhibits the growth of thyroid cancer cells, as well as increasing the expression of sodium/iodide symporter (NIS, encoded in humans by the *SLC5A5* gene), improving the prognosis of patients with thyroid cancer [355]. Lobeglitazone inhibits the migration and invasion of thyroid cancer due to the inhibition of the p38 MAPK pathway [356], whereas pioglitazone induces apoptosis through a PPAR-γ-independent pathway [357]. As described above, anti-cancer action may be caused by different mechanisms. For example, ciglitazone stimulates the expression of p21 and suppresses the Cyclin D1 by PPAR-γ independent pathway, whereas rosiglitazone acts through a PPAR-γ-dependent pathway [358]. There are also several other mechanisms of anti-cancer activity of TZDs [321]. The above-mentioned results obtained in several studies reveal the anti-tumor action of TZDs. On the other hand, there are also results revealing that TZDs show no anti-cancer action, or no association between use of TZDs, and there are results which suggest that the use of TZDs in diabetic patients increases risks for certain cancers. A systematic review and meta-analysis of 92 studies revealed that TZDs were associated with a lower risk of breast cancer (RR = 0.87; 95% CI = 0.80–0.95), lung cancer (RR = 0.77; 95% CI = 0.61–0.96), and liver cancer (RR = 0.83; 95% CI = 0.72–0.95) [30]. The results obtained in the case of pioglitazone, used in diabetic patients with bladder cancer, are different and sometimes controversial. In one study, pioglitazone revealed a modestly increased risk of bladder cancer development in diabetic patients (RR = 1.20; 95% CI = 1.07–1.34) [359,360], however, this association was not observed in the case of rosiglitazone (OR = 0.91; 95% CI = 0.71–0.1.16] [344]. In a large-scale prospective cohort study, researchers found no association between the use of pioglitazone and the risk of bladder cancer (HR = 1.06; 95% CI = 0.89–1.26), however, in the same cohort study, the researchers found an increased risk of bladder cancer disappearing after >4 years of this TZD use [361]. In the case of pioglitazone and rosiglitazone, for colorectal cancer, a slight inverse relationship was observed (RR = 0.93; 95% CI = 0.90–0.97), and stronger inverse relationship was observed for liver cancer (RR = 0.65; 95% CI = 0/48–0.89) [359]. The obtained results revealed that TZDs’ use carried no risk, or a potentially lower risk of some cancers [30]. Because these results are different, sometimes opposite, and controversial, the association between TZDs and cancer requires further investigations.

### 5.3. Sulfonylureas

Sulfonylureas (SUs) are still extensively used in the treatment of T2DM. These drugs act as insulin secretagogues [124]. Currently in use as antidiabetic drugs for the management of T2DM are second-generation sulfonylurea drugs, such as glibenclamide, gliclazide, glipizide, and glimepiride [321]. These drugs increase the release of insulin from pancreatic β-cells [362]. It was found that glilenclamide acts through sulfonylurea receptors which are expressed in pancreatic cells. Sulfonylurea receptors are subunits of adenosine triphosphate-sensitive potassium channels (K^+^ATP channels). These channels may be inhibited by the above-mentioned drug, causing cell depolarization, the opening of voltage-gated calcium channels, and then calcium influx into the cells, resulting in the secretion of insulin due to vesicle exocytosis [363]. The release of insulin into the blood stream facilitates the uptake of glucose into peripheral cells. 

Results obtained in previous studies revealed that the use of sulfonylureas in diabetic patients increases the incidence of cancer development and risk of cancer mortality [28,364], especially in pancreatic and breast cancer [365]. On the other hand, there are also some randomized controlled trials in which no statistical differences were observed in the risk of cancer development between the use of SUs and other treatments [366]. In a performed meta-analysis in which 24 metformin studies and 18 sulfonylureas studies were analyzed, the correlation between anti-diabetic drugs and cancer incidence [367] was investigated. The results obtained in this analysis showed an increased cancer risk only in cohort studies regarding sulfonylureas. On the other hand, case–control and randomized controlled trials failed to confirm this finding [368]. A retrospective cohort study revealed that diabetic patients who were treated with SUs had a higher risk of breast cancer death (HR = 1.49; 95% CI = 1.00–2.23] [365]. Most of these previous studies bore the burden of time-related biases, and the elimination of these biases means that no increased risk of cancer was detected [369]. Also, no evidence of any influence of glipizide or glimepiride on cancer growth has been observed [368]. Results on the association between use of SU drugs and cancer are different, controversial, and contradictory. An increased risk of cancer associated with the use of SUs was observed, while other observations noted a decreased risk of cancer or no change in the risks associated with using SUs [321]. 

There are different suggested anti-cancer mechanisms of sulfonylureas, such as the inhibition of multidrug-resistant proteins (MRPs) by glibenclamide in lung cancer cells. The treatment of non-small cell lung carcinoma by the mentioned drug suppresses cell growth, cell cycle progression, epithelial–mesenchymal transition, and cell migration [370]. It is suggested that the anti-cancer action of glibenclamide may be associated with closure of K^+^ATP channels expressed in cancer cells [368]. Observations performed on the gastric cancer cell line (MGC-803) revealed that glibenclamide can induce reactive oxygen species (ROS) generation and apoptosis of the cell. More detailed investigations showed that ROS generation activates the proapoptotic c-Jun N-terminal kinase and inhibits the activity of anti-apoptotic AKT kinase, resulting in the reduction in mitochondrial cytochrome c and apoptosis-inducing factor to the cytosol. The released compounds, in turn, may lead to caspase-dependent and independent apoptosis [371]. For more details, see [321].

However, although SU drugs have been used clinically for many years, their associations with cancer remain unclear [306]. Further and more extensive studies are required for a better understanding of the association between sulfonylureas drugs and cancer [124].

### 5.4. Sodium-Glucose Transporter-2 Inhibitors

Sodium-glucose transporter-2 (SGLT-2) inhibitors (SGLT-2I), such as empagliflozin, dapagliflozin, canagliflozin, and ertugliflozin, used in diabetic patients can lower the blood glucose due to the inhibition of renal glucose reabsorption and an increase its urinary excretion. This strategy reduces hyperglycemia. SGLT-2Is are relatively new in clinical practice. Therefore, the association between SGLT-2 inhibitors and the risk of cancer remains uncertain; however, based on a meta-analysis of randomized controlled trials, it is suggested that these drugs are not associated with an increased risk of cancer [27,372]. The results obtained in the mentioned meta-analysis showed no statistically significant association between overall cancer risk and using SGLT-2 inhibitors (OR = 1.14; 95% CI = 0.96–1.36) [27]. This suggestion was also proposed by other researchers [373,374]. There are also suggestions that these inhibitors may be useful for cancer therapy. For example, canagliflozin may protect against gastrointestinal cancers (OR = 0.15; 95% CI = 0.40–0.60) [27]. Also, preclinical studies revealed that canagliflozin has an anti-proliferative effect on liver cancer cells [375,376,377,378]. The results obtained by other authors revealed that these inhibitors may be useful for anticancer therapy, because these drugs increase cancer necrosis and, hence, induce cancer shrinkage [379]. As mentioned above, canaglioflozin protects against gastrointestinal cancers, in the case of bladder cancer, the risk increases (OR = 3.87; 95% CI = 1.48–10.08), and empagliflozin has the strongest association with bladder cancer (OR = 4.49; 95% CI = 1.21–16.73) [27], however, another observation revealed no association between empagliflozin and bladder cancer [380]. Also, another action of canagliglozin was found, which can inhibit cancer growth, causing the inhibition of complex I of the mitochondrial respiratory chain [381].

### 5.5. α-Glucosidase Inhibitors

The role of α-glucosidase inhibitors, such as acarbose, is the inhibition of glycosidase at the brush border of the small intestinal mucosa. The inhibition of glycosidase reduces the intestinal absorption of complex carbohydrates [382]. Acarbose has a low effect on reducing post-prandial glucose levels, as well as a very low risk of hypoglycemia [9]. Performed research revealed that acarbose may be used in anti-cancer therapy. Studies on mice with colon cancer and melanoma showed that acarbose significantly inhibits cancer growth and further enhances the therapeutic effect of anti-PD1 (programmed cell death receptor 1) [383]. A population-based observation revealed the risk of lung cancer in diabetic patients due to use of acarbose [384]. A dose-dependent reduction in the risk of colorectal cancer by approximately 27% in diabetic patients [385] was also detected. Acarbose reduces thyroid cancer metastasis [386]. Limited clinical trials mean that further large-sample multicenter studies are needed to verify the influence of acarbose on growth of thyroid cancer cells [185].

### 5.6. Incretin-Based Drugs

Incretin-based therapy, such as dipeptidyl peptidase-IV (DPP-VI) inhibitors and glucagon-like peptide-1 (GLP-1) receptor (GLP-1R) agonists, is increasingly used in T2DM. These drugs enhance or mimic the effect of gut-derived incretin hormones. They stimulate secretion of insulin, improve glycemic control inT2DM, and suppress postprandial levels of glucagon, as well as delay gastric emptying [232]. DPP-IV inhibitors inhibit the action of the enzyme that degrades endogenous GLP-1 and other peptides. The second kind of mentioned incretin binds to the GLP-1 receptor and shows agonist activity. Results taken from animal studies revealed that liraglutide, a GLP-1 receptor agonist, increases the risk of medullary thyroid cancer in rats and mice [387]. There was also an observation that suggests that GLP-1 receptor agonists decrease the risk for colorectal cancer patients with T2DM. An observation was performed on diabetic patients who had been prescribed anti-diabetic medications, where no prior anti-diabetic medication was used (drug naïve) and there was no prior CRC diagnosis. The effect of GLP-1 receptor agonists on the risk of CRC cancer was compared with diabetic patients who were treated with insulin, metformin, α-glucosidase inhibitors, DDP-IV inhibitors, SGLT-2 inhibitors, sulfonylureas, and thiazolidinediones [388]. The obtained results revealed that, in drug naïve patients, GLP-1 receptor agonists decreased the risk of CRC in comparison with insulin (HR = 0.56, 95% CI = 0.44–0.72), metformin (HR = 0.75; 95% CI = 0.58–0.97), SGLT-2 inhibitors, sulfonylureas, and thiazolidinediones. The cancer risk in patients treated with GLP-1R agonists was lower as compared to patients treated with α-glucosidase inhibitors and DPP-IV inhibitors; however, these differences were not statistically significant. It was observed also that the use of GLP-1 receptor agonists was associated with a lower risk of colorectal cancer in diabetic patients with obesity/overweight, compared to patients treated with insulin (HR = 0.50; 95% CI = 0.33–0.75), metformin (HR = 0.58; 95% CI = 0.38–0.89), and other anti-diabetics [388].

### 5.7. Insulin and Insulin Analog

All patients with T1DM require exogenous insulin therapy to prevent and treat hyperglycemia. Also, in many patients with T2DM, insulin treating hyperglycemia in the case of a progressive loss of β-cell function over time is required. Recombinant DNA technology enables the production of recombinant human insulin. There are several insulin formulations: short-acting human regular insulin (lispo, aspart, glusine), intermediate-acting human NPH insulin, the rapid-acting analog of human insulin, and the long-acting analog of human insulin (glargine) [232].

Noted observations suggest that patients with insulin therapy have a higher risk of malignancies when compared to patients with no insulin use [389], (RR = 1.39; 95% CI = 1.14–1.70) [390]. An increased risk of malignancies in diabetic patients with insulin and insulin analogs therapy was observed in the cases of cancers of the colorectum, breast pancreas, liver, kidney, stomach, and respiratory system [25,391]. On the other hand, there are also results suggesting a negative association between insulin and the risk of breast cancer (RR = 0.90; 95% CI = 0.82–0.98) and prostate cancer (RR = 0.74; 95% CI = 0.56–0.98), whereas in the case of liver cancer (RR = 1.74; 95% CI = 1.08–2.80) and pancreatic cancer (RR = 2.41; 95% CI = 1.08–5.36), a positive association with insulin use [30] was observed. The results obtained in a retrospective study revealed that the use of insulin and analogs, such as aspart, lispro, and glargine in diabetic patients, revealed a dose-dependent increased risk of cancer [34]. Animal studies revealed that use of insulin increases colonic epithelial tissue proliferation, resulting in colon cancer growth [392]. Positive associations between insulin therapy and cancer development were observed in the case of liver cancer (RR = 1.74; 95%CI = 1.08–2.80) and pancreatic cancer (RR = 2.41; 95% CI = 1.08–5.36) [18]. In other investigations, a performed meta-analysis revealed that insulin is associated with some cancers [25]. A comparison of insulin therapy to non-insulin therapy, for overall incidence, showed RR = 1.52; 95% CI = 1.16–2.00. This association was stronger for colorectal cancer (RR = 1.79; 95% CI = 1.36–2.36) and for pancreatic cancer (RR = 3.83; 95% CI = 1.43–10.23). Researchers also found that shorter durations of insulin therapy were associated with a higher risk of cancer development in comparison with longer insulin exposure [25]. An in vitro study performed on cell lines HCT-116 (colorectal cancer), PC-3 (pancreatic cancer), and MCF-7 (breast adenocarcinoma) in which these cancer cells were treated with insulin, insulin analogs, and IGF-1 showed interesting results [393]. Insulin analogs, unlike human insulin, showed IGF-1-like anti-apoptotic and mitogenic activation. IGF-1 was involved in cancer initiation and progression, while glargine stimulated phosphorylation of insulin and IGF-1 receptors [393]. Another in vitro study confirmed that insulin glargine activates IGF-1 receptors and also activates the MAPK pathway in MCF-7 cells and acts as mitogen in cells which have a high IGF-1/receptor/insulin receptor ratio, as observed in MCF-7 cells [394]. In humans, glargine is quickly converted into a mitogenic metabolite, therefore, a theoretical risk does not appear to be present [84]. There are also other observations, but the results are different and inconsistent. For example, a meta-analysis of 16 cohort and 3 case–control studies [395] was performed. Only 15 of these 19 studies had a measurement for any cancer. The obtained results showed that, in 13 of 15 studies, there were no associations between insulin glargine and detemir and any cancer. Also, measurement for breast cancer was performed in only 13 of the 19 studies. In 4 of the 13 studies, an increased risk of breast cancer associated with insulin glargine therapy was observed. Based on the quality assessment, randomized controlled trials did not reveal an increased risk of cancer due to the use of insulin [395]. Therefore, the authors suggest that the results obtained in the performed meta-analysis were too inconsistent to definitively determine a risk between analog basal insulin and cancer [395]. According to the researchers’ suggestions, a conclusion about the risk of cancer development in patients treated with a long-acting insulin analog cannot be made, because all of the relevant studies have had methodological problems that limited the researchers’ ability to draw conclusions [396]. It is suggested that studies should include insulin dose, length of treatment, and duration of diabetes mellitus, as necessary in investigations of the dependence of insulin analogs therapy on cancer development [84]. Based on the results obtained in another study, a neutral effect of insulin glargine on cancers [397,398] is postulated. Also, other research confirmed no increased risk for the development of any cancer due to use of insulin glargine or detemir, in comparison with human insulin [399]. It was confirmed also that insulin glargine, administered in humans, is rapidly transformed into active metabolites. These metabolites have the some mitogenic properties, as observed in the case of human insulin [400]. Therefore, it is suggested that there is no strong evidence that the use of insulin analogs is associated with an increased risk of cancer [83,306]. Additionally, several different studies, such as randomized controlled trial studies, cohort studies, and systematic reviews, confirmed the suggestion that insulin analog therapy is not associated with the risk of cancer overall and some site-specific cancers [9,34,401].

## 6. Future Direction and Perspective

Both diabetes mellitus and cancer are becoming more common worldwide. Results obtained in several studies revealed the association between diabetes mellitus and carcinogenesis. There are a few mechanisms suggested for this dependence, but the underlying mechanism is not well known. The association for both diseases is complicated. Therefore, further studies, such as clinical observations, meta-analyses, and population-based cohort studies, are needed to determine whether the association is caused or influenced by other factors. The obtained results may help to use primary prevention and early detection, as well as effective therapeutic measures. The establishment of national programs and national policies, as well as databases of patients with cancer and diabetes, and an increased focus on health services are all important preventive measures for diabetes and cancer and for better outcomes in the prevention of diabetes and cancer.

In obese patients and those with T2DM, several factors, such as hyperinsulinemia, hyperglycemia, dyslipidemia, adipokines, and cytokines, may contribute to the development and progression of cancer. The role of the gut microbiome as one of the factors influencing diabetes mellitus and cancer [53] should also be investigated.

The relationship between diabetes and cancer is not established. Some antidiabetic drugs may increase the risk of cancer development and progression, whereas with others, it may decrease. On the other hand, some anticancer therapies may induce hyperglycemia, hyperinsulinemia, and diabetes mellitus, whereas other therapies influence only cancer, without the stimulation of diabetes. The above-mentioned directions of investigations may have a positive effect on the battle against diabetes mellitus and cancer [402,403,404,405].

## 7. Conclusions

DM and cancer are certainly becoming more common worldwide. Currently, there is good evidence in epidemiological studies of a high prevalence of cancer and DM. The association between these two diseases is complicated, and this association has been well demonstrated in many studies. It was found that the rapidly growing incidence of DM and cancer worldwide is due to changes in lifestyle. The obtained results suggest that the promotion of a healthy lifestyle may reduce the risk of developing cancer and DM.

The relationship and exact mechanism between different antidiabetic drugs showed an anticancer effect. On the other hand, several other antidiabetic drugs may increase the cancer risk. Fully understanding of these relationships remains an important area of investigations with potentially vital implications for the future of medicine. Heterogeneity characterizes these both diseases, diabetes mellitus and cancer, meaning that studies on the association between these two diseases are not easy to carry out. Also, the possible suggested different mechanisms which cause increased cancer incidence and mortality in diabetic patients shows that it is difficult to accurately define the aims and the recruitment criteria for these studies.

DM represents a risk factor for cancer. Thus, cancer should be screened in routine diabetes assessments. Unfortunately, there are many different and sometimes controversial results. There are also many still-unanswered questions, such as are mechanisms of the association between both types of DM and cancer specific for type of DM, or the same, which type of DM is more associated with a particular type of cancer, and how to decrease the risk of developing cancer in diabetic patients. These problems, as well as many others, need further investigations.

In our review, we have summarized the most recent studies regarding the relationship between DM and cancer, as well as presenting the risk of drug therapy. We hope that this review will help both clinicians and researchers to design new experiments for obtaining more successive forms of therapy.

## Figures and Tables

**Figure 1 ijms-25-07476-f001:**
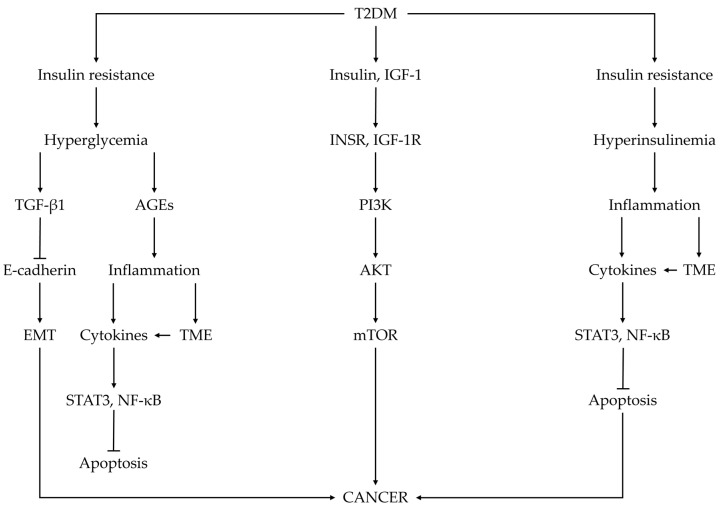
Association of type 2 diabetes mellitus with cancer, for example, pancreatic cancer. The role of insulin and insulin-like growth factor 1 intracellular signaling pathway and insulin resistance on development of cancer. AGEs—advanced glycation end products, AKT—protein kinase B, EMT—epithelial–mesenchymal transition, IGF-1—insulin-like growth factor-1, IGF-1R—insulin-like growth factor-1 receptor, INSR—insulin receptor, mTOR—mammalian target of rapamycin, NF-κB—nuclear factor kappa B, PI3K—phosphatidylinositol 3 kinase, STAT3—signal transducer and activator of transcription 3, T2DM—type 2 diabetes mellitus, TGF-β1—transforming growth factor-β1, and TME—tumor microenvironment [14,50,87,111,223,224].

**Table 1 ijms-25-07476-t001:** Mortality of patients with T2DM in dependence on cancer and sex. 95% CI—95% Confidence interval.

Cancer Type	Random Effects (95% CI)	Ref.
Oral	1.41 (1.16–1.72)	[8]
Gastric	1.28 (0.93–1.76); 1.29 (1.04–1.59)	[8,124]
Colorectal	1.20 (1.03–1.40); 1.20 (1.03–1.40); 1.12 (1.01–1.24)	[8,124,150]
Hepatocellular carcinoma	2.43 (1.67–3.55); 2.43 (1.67–3.55);	[8,124]
Gallbladder	1.30 (1.07–1.59)	[8]
Breast	1.24 (0.95–1.62); 1.24 (0.95–1.62); 1.57 (1.23–2.01)	[8,124,193]
Endometrial	1.32 (1.13–1.55); 1.23 (0.78–1.93);	[8,124]
Kidney	1.16 (1.01–1.33)	[124]
Cancer mortality in women		
Gastric	1.24 (0.95–1.63)	[53]
Colon	1.18 (1.04–1.33)	[53]
Liver or intrahepatic bile duct	1.40 (1.05–1.86)	[53]
Pancreas	1.31 (1.14–1.51)	[53]
Breast	1.16 (1.03–1.29)	[53]
Endometrial	1.33 (1.08–1.65)	[53]
Cervix	1.47 (1.19–3.03)	[53]
Kidney and urinary organs	1.15 (0.88–1.52)	[53]
Cancer mortality in men		
Oral cavity and pharynx	1.44 (1.07–1.94)	[53]
Colon	1.15 (1.03–1.29)	[53]
Liver or intrahepatic bile duct	2.26 (1.89–2.70)	[53]
Pancreas	1.40 (1.23–1.59)	[53]
Bladder	1.22 (1.01–1.47)	[53]
Breast	4.20 (2.20–8.04)	[53]
Prostate	0.88 (0.79–0.97)	[53]

**Table 2 ijms-25-07476-t002:** Specific cancer risk in diabetes. SRR—standardized rate ratio; RR—relative risk; and OR—odds ratio.

Cancer Type	Risk Estimates (95% CI)	Ref.
Liver	SRR = 2.01 (1.61–2.51)	[36]
Pancreas	RR = 1.94 (1.66–2.27)	[95]
Colorectal	RR = 1.26 (1.20–1.31)	[154]
Esophagus	SSR = 1.30 (1.12–1.50)	[123]
Kidney	RR = 1.42 (1.06–1.91)	[159]
Bladder	RR = 1.24 (1.08–1.42); 1.35 (1.17–1.58)	[36,124]
Breast	RR = 1.20 (1.12–1.28); 1.25 (1.20–1.29)	[8,192]
Endometrial	RR = 1.65 (1.50–1.81); 1.89 (1.46–2.45); 2.22 (1.80–2.74); 2.10 (1.75–2.53); OR = 1.70 (1.2–2.3)	[8,31,198,201,202]
Blood	Non-Hodgkin’s Lymphoma OR = 1.22 (1.07–1.39); Leukemia OR = 1.22 (1.03–1.44); Myeloma OR = 1.22 (0.98–1.53)	[184]

## Data Availability

Not applicable.

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
