# Peer review of "Associations between Diabetes Mellitus and Selected Cancers"

_ijms, 2024, doi:10.3390/ijms25137476_

Round 1

Reviewer 1 Report

Comments and Suggestions for Authors

Dear Editor

Many thanks for considering me a potential reviewer for the said article entitle; Associations between Diabetes Mellitus and Selected Cancers. This review is well structured and written, however, here are some queries (IMPORTANT) and minor corrections (suggestions) that must be taken into consideration.

My observations are as follow;

Major Comments

1.     For the moment, I just copy your tittle and search in the google and found some interesting articles like, (https://doi.org/10.1677/ERC-09-0087). You also did a nice effort, however, please add some figures and/or pathways, soo it will increase the readers interest.

2.     Authors need to pay attention to cite a proper source when using a claim (please check whole manuscript), for example,

·       Line 199-120; Several studies revealed that T2DM is an independent risk factor that may influence risk cancers, such as hepatic, pancreatic, bladder, endometrial, colorectal and breast cancer.

·       Line 133; Pancreatic cancer (PC) is one of the most lethal malignant disease,

·       Line 190; Hepatocellular carcinoma (HCC) is the most common type of primary liver cancer. It is the fifth most common cancer in men and seventh one in women,

·       Line 222; Esophageal carcinoma (EC) is the sixth most common cause of death. It is also, as previous mentioned cancers, associated with T2DM,

·       Analysis performed in 2011 (line-225) no citation.

3.     Table 1; what is SRR, PR and so on… CI and Ref…... The authors are advised to write its full form in the legends, although, there is full form in the text but here it causes confusion,

4.     Line 558; ‘Underlying mechanisms of associations between diabetes mellitus and cancer’ I am not sure, why authors skipping citations?.

Minor comments  

1.      8% - 18% replace with 8-18%,

2.     WHERE IS CITATION/SOURCE? ‘T1DM is characterized by profound absolute by insulin deficiency caused by autoimmune destruction of the pancreatic β-cells. This disease requires exogenous administration of insulin. T1DM has not been linked with the increased risk of cancer. The knowledge about the risk of cancer in patients with T1DM is poor. The association of T1DM with cancer is not well described’.

3.     WHERE IS CITATION/SOURCE? ‘T2DM is metabolic disease due to insulin resistance of peripheral tissues and cells, resulting increased blood glucose levels (hyperglycemia). Chronically, prolonged increased blood glucose levels may cause end organ damage, such as nephropathy or retinopathy. Elevated level of circulating glucose stimulates increased secretion of insulin by pancreatic β-cells to obtain normoglycemia. Both hyperglycemia and hyperinsulinemia may be associated with development of cancer’.

4.     Pay attention to the references number,

Comments on the Quality of English Language

Dear Authors, 

It's very difficult to understand the text main idea, please pay attention to the mention observations and I will strongly recommend to do extensive english editing by a professional in the field.

Thanks

Author Response

Dear Reviewer

Reviewer 2 Report

Comments and Suggestions for Authors

This review discusses the link between diabetes and cancer. This topic is very important. But some papers have been published in this context.

  1. Various risk factors, like intrinsic and extrinsic factors of cancer, should be provided.
  2. How can diabetes lead to cancer? Focus on gene levels and DNA damage.
  3. Signaling cascades become altered in diabetic patients, which may contribute to tumor formation. It is advised to write a paragraph on such signaling molecules and pathways.
  4. The title is misleading because the mechanisms are partially included.
  1. Various risk factors causing cancer in diabetic patients are not provided.
  2. The authors should discuss a link between glycation and advanced glycation endproducts in cancer. Include the following reference.

Advanced glycation endproducts have been reported to be found in various human tumors. It is very interesting to note that a higher level of AGEs accumulation is reported to be associated with malignant tissues, such as prostate cancer, compared to benign tumors.

A review on mechanism of inhibition of advanced glycation end products formation by plant derived polyphenolic compounds. Mol. Biol. Rep. 2021, 48, 787–805.

  1. The authors should make a figure showing multiple kinds of cancers commonly linked with diabetes.
  2. Necrosis and apoptosis and their implications for diabetic foot cancer should be discussed.
  3. Why does insulin therapy increase the incidence of cancer?
  4. The authors should make a list of anticancer drugs that may cause hyperglycemia and insulin resistance and their mechanisms.
  5. The introduction is so superficial, with insufficient reference citations. This part should be developed more and not be restricted to the definition and types of diabetes.
  6. In addition, please make a new paragraph about the perspective.
  7. What is the novelty of your work compared to others?
  8. The conclusion section lacks a concise summary of the key findings and their implications. A more focused conclusion would enhance the overall impact of the study.

Comments on the Quality of English Language

The syntax, commas, space and grammatical errors should be checked.

Author Response

Dear Reviewer

Reviewer 3 Report

Comments and Suggestions for Authors

This manuscript presented a thorough review with an emphasis on the connections between diabetes and different types of cancer. The review emphasizes the long-established link between diabetes and cancer, pointing out that 8% to 18% of diabetes patients also have cancer diagnoses at the same time. The review also covers how diabetes can be exacerbated by cancer and its treatments, which can impact treatment decisions due to complications from diabetes. This article essentially summarizes the most recent data and theories regarding the mechanisms relating diabetes to cancer, highlighting the need for more intensive research. Still, some arguments seem to be more compelling than others, particularly when it comes to potentially preventive measures that target diabetes mitigation to lower the risk of cancer. A list of particular remarks is also provided below. 

1.     Diabetes can be further classified into a wide range of types clinically. Although it appears to be limited to paragraph 2.1, the writers have given a brief synopsis of T1D. Actually, the topic of T2D takes up a great deal of the entire article. While Type 2 Diabetes (T2D) is the most prevalent type of the disease, I believe that other types of diabetes should also be discussed and their similarities and differences with T2D should be highlighted if the topic is to be appropriate. 

2.     In terms of technical details, the authors talked about hyperglycemia, inflammatory cytokines, insulin resistance, hyperinsulinemia, and other potential causes like genetics. These talks don't seem to cover enough ground, though, and the material could be expanded upon and covered in greater detail. For instance, obesity is a significant risk factor for type 2 diabetes but has not received much attention. Furthermore, very little information regarding genetic factors is covered. I believe that this aspect needs to be strengthened. 

3.     There doesn't seem to be much discussion about anti-diabetic medications. The manuscript might examine the relationship between cancer risk and various diabetes management techniques. For example, the impact of insulin versus other hypoglycemic agents on the risk of cancer could be compared. Furthermore, the way that diabetes and cancer are treated may change how they affect one another. It would also be helpful to talk about how the most recent treatment developments (like SGLT2i or GLP-1RA) might change earlier conclusions regarding the association between diabetes and cancer. 

4.     Most likely, observational data—which can prove correlations but not causality—is what the review is based on. To determine whether diabetes directly causes cancer to develop or if the two conditions share risk factors like age and race, more prospective and longitudinal studies are needed. 

5.     To make it easier for readers to compare the information in each table, I propose that we think about grouping the different cancer types according to decreasing power. 

6.     Tab. 1: More comorbid conditions appear to have an impact on the mortality of DM patients. Is it feasible to think about displaying the incidence rates of multiple cancers at the same time? 

7.     Minors: Can the abstract contain a succinct summary of this study's findings? 

Comments on the Quality of English Language

Minor editing of English language required.

Author Response

Dear Reviewer

Please dee the attachment.

Reviewer 4 Report

Comments and Suggestions for Authors

This review describes the relationship between various cancers and diabetes, and I think it has great clinical significance. However, the authors need to revise the following points.

1.    The latest data is required to write about cancer, which is the leading cause of death in some developed countries, but some references are too outdated (ex. references 1 and 45).

2.    It would be helpful to clearly list cancers that are not affected by diabetes or whose effects are currently unknown.

3.    It is also necessary to mention that some antidiabetic drugs, such as metformin and SGLT2 inhibitors, and drugs for treating diabetic complications, such as aspirin and statins, have been reported to have anticancer effects.

4.    There are parts where the same expressions and phrases are repeatedly used, so careful and easy-to-read English proofreading is required.

Author Response

Dear Reviewer

Round 2

Reviewer 1 Report

Comments and Suggestions for Authors

Dear Authors/Editors,

Many thanks for the consideration.

I think the author has successfully addressed my queries and suggestions, hence the manuscript is much more refined. 

Furthermore, I have no concern for the onward step.

Kind regards,

Comments on the Quality of English Language

Dear Authors/Editors,

The article is much refine, however, would be nice if minor English editing is done by a professional in the field.

Author Response

Dear Reviewer

Thank you very much for your opinion. The proofreading of this manuscript was done previous by professional editor. But after your suggestion, he again looked at this manuscript and included few, minor, changes.

Reviewer 2 Report

Comments and Suggestions for Authors

The authors have provided a nicely detailed and thorough response to the comments from the previous review and have addressed my major concerns. The manuscript now reads with greater focus and clarity. This revision has significantly improved the manuscript. I enjoyed reading it.

Therefore, I think that the manuscript is adequately revised and it can be recommended for publication.

Author Response

Dear Reviewer

Thank you very much for your opinion and decision.

Reviewer 4 Report

Comments and Suggestions for Authors

Figure 1 should be created more carefully and neatly.

Author Response

Dear Reviewer

Thank you very much for your opinion and suggestion. Unfortunately, it is problem with Fig. 1. Its primary version was bigger but after reduction, there are problems with lines and words (place is too small). Therefore, I can't change it, however I worked with this problem, but without result.